# Balanced SET levels favor the correct enhancer repertoire during cell fate acquisition

Mattia Zaghi[1], Federica Banfi[1,2], Luca Massimino[3], Monica Volpin [4], Edoardo Bellini [1], Simone Brusco[1,2], Ivan Merelli[5], Cristiana Barone[6], Michela Bruni[7], Linda Bossini[1], Luigi Antonio Lamparelli[3], Laura Pintado[1], Deborah D'Aliberti[6], Silvia Spinelli[6], Luca Mologni [6], Gaia Colasante [1], Federica Ungaro[3], Jean-Michel Cioni[7], Emanuele Azzoni [6], Rocco Piazza [6], Eugenio Montini [4], Vania Broccoli [1,2] & Alessandro Sessa [1] ✉

Within the chromatin, distal elements interact with promoters to regulate specific transcriptional programs. Histone acetylation, interfering with the net charges of the nucleosomes, is a key player in this regulation. Here, we report that the oncoprotein SET is a critical determinant for the levels of histone acetylation within enhancers. We disclose that a condition in which SET is accumulated, the severe Schinzel-Giedion Syndrome (SGS), is characterized by a failure in the usage of the distal regulatory regions typically employed during fate commitment. This is accompanied by the usage of alternative enhancers leading to a massive rewiring of the distal control of the gene transcription. This represents a (mal)adaptive mechanism that, on one side, allows to achieve a certain degree of differentiation, while on the other affects the fine and corrected maturation of the cells. Thus, we propose the differential *in cis*-regulation as a contributing factor to the pathological basis of SGS and possibly other the SET-related disorders in humans.

Throughout life and particularly during ontogeny, a variety of transcriptional programs grant the right and timely implementation of stemness potential, fate commitment, proper differentiation and adult function within every organ[1–4]. These cell-specific programs are controlled by dynamic changes in chromatin regulation, including the activity of distal noncoding DNA sequences, or enhancers, that interact with gene promoters[5–9]. For example, both the activation and repression of key lineage genes' outcomes can be primed by chromatin changes[10–12]. This directly implies the existence of convergent and divergent epigenetic features among different cells/cell states that in turn may share or not special requirements and specific vulnerabilities.

Since alterations in chromatin state have been documented in many human conditions, including neurodevelopmental disorders, rare genetic syndromes, and multiple cancers[13–15], a better knowledge of the enhancer activity and its burden in pathogenesis is required. Indeed, despite different mechanisms and players have been associated with it, including nuclear spatial organization, the role of transcription factor (TF) function, chromatin regulators, and nucleosome remodeling complexes, the fine-tuning and the specific usage of the different enhancer repertoires are only partially known[7,16,17].

Histone acetylation is a key epigenetic marker of active chromatin domains, including functional enhancer-promoter pairs, for its

[1]Stem Cell and Neurogenesis Unit, Division of Neuroscience, IRCCS San Raffaele Scientific Institute, 20132 Milan, Italy. [2]CNR Institute of Neuroscience, 20129 Milan, Italy. [3]Esperimental Gastroenterology Unit, Division of Immunology, IRCCS San Raffaele Scientific Institute, 20132 Milan, Italy. [4]San Raffaele Telethon Institute for Gene Therapy (SR-Tiget); IRCCS, San Raffaele Scientific Institute, 20132 Milan, Italy. [5]CNR Institute of Biomedical Technologies, 20090 Segrate, Italy. [6]School of Medicine and Surgery, University of Milano-Bicocca, 20900 Monza, Italy. [7]RNA biology of the Neuron Unit, Division of Neuroscience, IRCCS San Raffaele Scientific Institute, 20132 Milan, Italy. ✉e-mail: sessa.alessandro@hsr.it

capability to make the chromatin accessible[18–20]. For instance, high levels of acetylation in H3 lysine 27 (H3K27ac), 9 (H3K9ac), and 4 (H3K4ac) correlate with active transcription so that defects in dedicated histone acetyltransferases, e.g., P300/CBP and/or the relative deacetylases may heavily impact transcriptional programs and consequently development and homeostasis of different organs resulting in severe diseases[21].

Besides its documented role as an inhibitor of PP2A phosphatase[22], the oncoprotein SET has been found as part of the histone chaperone complex INHAT that blocks the activity of histone acetyltransferases[23–25]. This is possibly due to the stickiness of the acidic portion of SET for unacetylated lysine-rich domains that generates physical hindrance on histones[26]. Therefore, SET, as an inhibitor of acetyl mark independent of the DNA sequence, may represent an interesting entry point to study chromatin remodeling genome-wide. However, very little is known about the general consequences at the epigenomic and transcriptomic levels of the quantitative alterations of SET that have been documented in several diseases, e.g., cancer, myeloproliferative diseases, intellectual disability and Schinzel-Giedion syndrome (SGS)[27–34]. The latter is an ultra-rare syndrome characterized by severe developmental delay, progressive brain atrophy, other congenital malformations, and frequent seizures that do not respond to any known medication[33,35]. Consequently, affected children have a high mortality rate in their first years of life[36]. SGS is caused by mutations in *SETBP1* gene that results in the accumulation of the corresponding encoded SET binding protein 1 in these patients, further leading to the downstream accumulation of SET itself[33,37].

This study addresses whether and how SET controls the chromatin landscape to ensure physiological cell function and development. To do so, we have experimentally manipulated SET levels in different organisms, cell types and developmental trajectories, employing specifically SGS models, in which SET is endogenously accumulated, as paradigmatic cases. Genome-wide and single-cell epigenomic, 3D conformation, and transcriptomic studies have been used to decode the chromatin response to SET function as one of the possible pathological contributors to the SGS. These analyses allow us to elucidate the role of SET in securing the correct enhancer accessibility and bending to cognate promoters, required for fate-specific transcriptional programs. Our data illustrates that SET abnormal binding induces a chromatin rewiring of the distal control of gene transcription that employs different arrays of putative enhancers to achieve physiological transitions, eventually resulting in pathological phenotypes.

## Results

### SET levels influence histone acetylation in multiple models
The accumulation of SET characterizes several human diseases[29–34] leaving the hypothesis that alteration of histone acetylation may contribute to the pathological traits[34]. To directly assess the outcome of high levels of SET, we generated a stable line of human induced pluripotent stem cells (iPSCs) carrying the SET transgene under the control of a tetracycline-inducible promoter (Supplementary Fig. 1a). Upon doxycycline administration, the exogenous SET rapidly accumulated in both iPSCs and iPSC-derived neural precursor cells (NPCs) (Supplementary Fig. 1b, c). Both cell types showed a decreased in histone H3 pan-acetylation over the time of SET accumulation (Supplementary Fig. 1b, c). Since SET is evolutionarily conserved[38], we next investigated whether SET alters histone acetylation in non-mammalian species. Zebrafish embryos were injected with mRNA encoding the human *SET* or GFP as control, and analysis at 72 h post fertilization (h.p.f) revealed a significant decrease in the global levels of histone acetylation in *SET*-expressing embryos compared to control (Supplementary Fig. 1d).

Then, we moved towards a more (patho)physiological context using iPSC-derived NPCs from two independent SGS patients (two different *SETBP1* degron mutations: D868N and I871T), which are

characterized by a high level of SET as a secondary effect to SETBP1 accumulation[39] (Supplementary Fig. 1e, f). Reciprocal co-immunoprecipitation assays confirmed that SET and histone H3 interact, and indicated an increased amount of H3-bound SET in mutant NPCs, in line with SET accumulation in SGS (Fig. 1a). Accordingly, we revealed hypo-acetylation also in SGS NPCs compared to the CRISPR corrected isogenic controls (D868D and I871I)[39] in several lysine residues belonging to both, H3 and H4 histones (Fig. 1b, c).

To collect molecular details, we performed chromatin immuno-precipitation followed by sequencing (ChIP-seq) for H3K27ac and SET in SGS NPCs and relative isogenic controls. We revealed that the regions that are physiologically acetylated experienced a reduction in H3K27ac levels, specifically regions that are weakly decorated (Fig. 1d, Supplementary Data 1). As histone acetylation is intimately connected with chromatin accessibility[40–42], we performed assays for transposon accessible chromatin followed by sequencing (ATAC-seq)[43] to investigate whether SET accumulation leads to chromatin compaction. ATAC-seq datasets showed correlation with H3K27ac in both D868D and D868N NPCs as well as comparable correlations with RNA levels of associated genes (by proximity) (Supplementary Fig. 1g). The degree of correlation between control samples and between mutant samples of was highly similar, suggesting for a comparable effect of SGS mutations in each donor genotype (Supplementary Fig. 1g). Principal component analysis (PCA) demonstrated a remarkable similarity among SGS NPC samples at the level chromatin accessibility; likewise, their isogenic controls showed a strong tendency to cluster together and distant from SGS counterpart (Supplementary Fig. 1h). To determine the relevance of the accessible regions identified with our experiments, we performed an overlap analysis using previously classified chromatin states from available cell types and tissues[44]. The ATAC peaks from both D868D and D868N NPCs highly overlapped with enhancers and promoters accessible in brain tissues and ESC-derived NPCs while only to a lesser extent with cells and organ of different origin (Supplementary Fig. 1i, Supplementary Data 2). Looking at the regions that are normally acetylated in control NPCs, we found a slight decrease of ATAC signal (Fig. 1e) and a strengthen of SET binding (Fig. 1f) in SGS NPCs by ChIP-seq analysis. At the genome-wide level, all SET^high conditions examined were featured by consistently less called ATAC-seq peaks compared to the relative controls (Supplementary Fig. 1j), while all pairwise comparisons were similar in term of accessibility of the identified peaks (Fig. 1g, i, Supplementary Fig. 1k–m). Taken together these data suggest that SET accumulation could lead to specific defects associated with certain susceptible chromatin loci, instead of a pervasive defect.

### SET regulates chromatin accessibility in distal regulatory regions
To properly evaluate differences in chromatin accessibility, we analyzed the effect of SET accumulation in those regions that are physiologically open in control NPCs. We observed a tendency to chromatin closure in the SET^high conditions, both SGS NPCs, iPSCs, and NPCs overexpressing SET, and in the zebrafish, embryos injected with SET mRNA (Fig. 2a, Supplementary Fig. 2a–d). Subsequent K-means clustering analysis revealed three subgroups of regions (see Methods), with the third cluster in each analyzed pair, containing the modestly accessible regions, that were the most affected by SET accumulation becoming more closed (Fig. 2a, b, Supplementary Fig. 2a–d). The same DNA genomic regions displayed, at least in the D868 NPC pair, a strong H3K27 acetylation loss which was instead not or minimally detectable in the peaks of the other clusters (Fig. 2b).

The SET-sensitive regions in cluster 3 are mapped largely in introns and intergenic regions while clusters 1 and 2 are enriched in peaks proximal to gene promoters, in all the models analyzed (Fig. 2b, Supplementary Fig. 2a–d, and Supplementary Data 3). By using chromatin states from available neural cells and tissues[44] a specific overlap

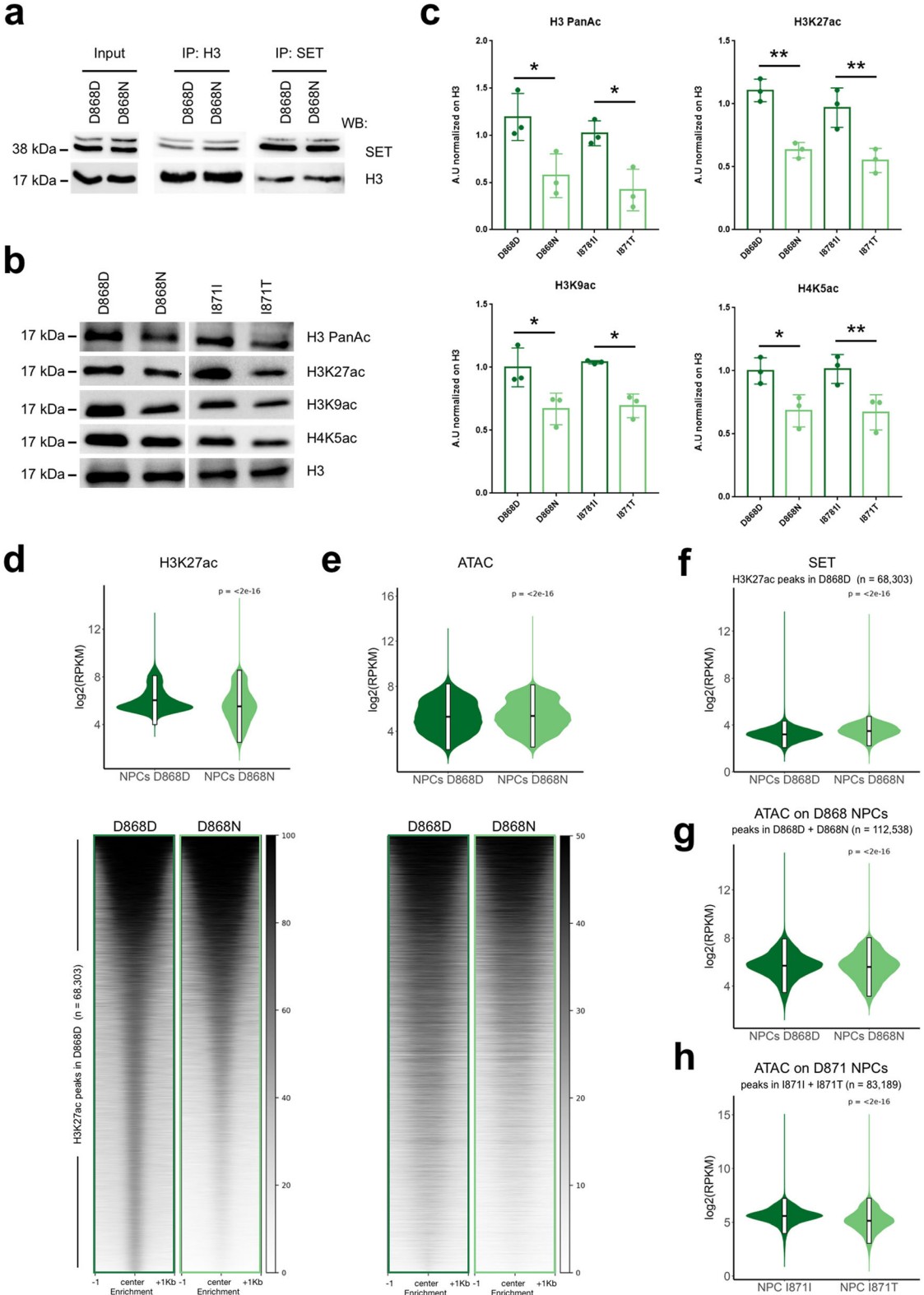

analysis of the peaks of the three subgroups, confirmed that cluster 3 contains a high number of distal regulatory regions, or putative enhancers, that are active in brain cells (Supplementary Fig. 3a, b). In contrast, the promoters are confined in the first two clusters (Supplementary Fig. 3a, b). The average expression of the genes associated to ATAC-seq peaks by proximity resulted relatively stronger in cluster 1 compared to the others (Supplementary Fig. 3c) and in line with the previous observations, the distance of the peaks to the nearest TSS is

higher in cluster 3 (Supplementary Fig. 3d). Performing motif analysis, we identified signatures of transcription factors (TFs) whose function may be modulated by these accessibility changes. In the cluster 3, we report the presence of CTCF sites as well as TFs with key roles in NPCs, e.g., SOX2, NEUROG2, BRN1, and members of the AP1 complex that typically binds enhancers in this cell type[45] (Fig. 2c, Supplementary Data 4). Conversely, cluster 1 peaks, which are equally accessible in both genotypes, are enriched for TFs usually found in promoters, such

**Fig. 1 | SET overexpression alters histone acetylation. a** SET, H3 Immunoprecipitation and Western blot analysis of SGS (D868N) and control (D868D) NPCs. **b**, **c** Western blot for: H3Pan, H3K27, H3K9, H4K5 acetylation in SGS (D868N) and control (D868D) NPCs (**b**) and relative quantification (**c**). H3 as loading control (Statistical analysis one-way ANOVA and one-sided Tuckey multiple comparison test. H3PanAc: D868DvsD868N *$P$ = 0.0296; D868DvsI871I $P$ = 0.7582; D868DvsI871T **$P$ = 0.0092; I871IvsI871T *$P$ = 0.0349; I871IvsD868N $P$ = 0.1191; D868NvsI871T $P$ = 0.8191. H3K27ac: D868DvsD868N **$P$ = 0.0027; D868DvsI871I $P$ = 0.4467; D868DvsI871T ***$P$ = 0.0010; I871IvsI871T **$P$ = 0.0058; I871IvsD868N *$P$ = 0.0197; D868NvsI871T $P$ = 0.7858. H3K9ac: D868DvsD868N *$P$ = 0.0260; D868DvsI871I $P$ = 0.9659; D868DvsI871T **$P$ = 0.0092; I871IvsI871T *$P$ = 0.0349; I871IvsD868N $P$ = 0.1191; D868NvsI871T $P$ = 0.9925. H4K5ac: D868DvsD868N *$P$ = 0.0519; D868DvsI871I $P$ = 0.9988; D868DvsI871T *$P$ = 0.0437; I871IvsI871T **$P$ = 0.0355; I871IvsD868N *$P$ = 0.0422; D868NvsI871T $P$ = 0.9993, $n$ = 3 independent experiments. Data are presented as mean values +/− SEM. **d**, **e** Heatmap (see methods) and violin plots of normalized signal of H3K27ac ChIP-seq and ATAC-seq in H3K27ac control peaks (D868D) NPCs, $n$ = 68,303 (Violin plot statistic (**d**) two-sided Wilcoxon-test $P$ < 2e-16. Boxplot with 25–75th percentiles, mean, and whiskers of minima to maxima; violin plot statistic (**e**), Wilcoxon-test $P$ < 2e-16. Boxplot with 25–75th percentiles, mean, and whiskers of minima to maxima). **f** Violin plots of normalized signal of SET ChIP-seq in significant H3K27ac ChIP-seq peaks in control (D868D) NPCs ($n$ = 68,303, statistic two-sided Wilcoxon-test $P$ < 2e-16. Boxplot with 25–75th percentiles, mean, and whiskers of minima to maxima). **g** Violin plots of normalized signal of ATAC-seq in significant peaks of NPCs control (D868D) and SGS (D868N) ($n$ = 112,538, statistic two-sided Wilcoxon-test $P$ < 2e-16. Boxplot with 25–75th percentiles, mean, and whiskers of minima to maxima) and **h** in significant peaks of NPCs control (I871I) and SGS (I871T), $n$ = 83,189 (statistic two-sided Wilcoxon-test $P$ < 2e-16. Boxplot with 25–75th percentiles, mean, and whiskers of minima to maxima).

as the SP family[46,47] (Supplementary Fig. 3e, Supplementary Data 4). Notably, we found high similarity in the comparison of chromatin states considering only the distal peaks (both intronic and intergenic) of the 1st and 2nd clusters (8426 putative enhancers) and those of the 3rd cluster (21,201 putative enhancers) (Supplementary Fig. 3f), suggesting that the SGS condition affects several but not all distal regulatory elements.

Thus far, SET accumulation seems to interfere with accessibility, mostly at putative enhancer regions rather than promoter proximal regions. Therefore, we wondered whether arrays of enhancers in close genomic proximity, known as super-enhancers (SEs)[48–50], could be affected as well. First, we identified 1387 SEs across control NPCs using the total background-subtracted H3K27ac ChIP-seq signal (Supplementary Fig. 4a, Supplementary Data 5). SEs were defined as the regions where the occupancy signal began to increase exponentially (Supplementary Fig. 4a). Mean value of H3K27ac along the SEs and the mean of ATAC-seq signal in peaks associated with SEs were not different between D868D and D868N NPCs (Fig. 2d). However, considering only the SEs with at least the 50% of associated ATAC peaks belonging to the cluster 3 ($n$ = 611), a decrease in both H3K27ac and chromatin accessibility was found (Fig. 2e). The same analysis for SE mainly associated to the other clusters revealed no differences in H3K27ac and even a higher chromatin accessibility in mutant than in control NPCs (Supplementary Fig. 4b, c).

We next investigated the identified differentially accessible regions from the neural developmental perspective. To do this, we compiled an open chromatin atlas between the PSC and the NPC stages by merging the ATAC peaks found in the two cell types (WT condition, 121,440 unique peaks), ranked them based on the fold change between NPC and PSC, and divided into four quartiles with the 1st containing PSC-specific accessible chromatin, and the 4th being NPC-specific (Fig. 2f, Supplementary Data 3). Interestingly, the larger loss of accessibility between D868D and D868N NPCs is located inside the 4th quartile (Fig. 2f, e). This data indicates that the accumulation of SET has a higher burden on regulatory regions that need to be opened during cell development and to a greater extend compared on those equally accessible in the two cell types.

These findings suggest that high levels of SET impair the opening of most distal regulatory regions, including several SEs, specific to neural fate commitment.

**SGS NPCs display rearranged chromatin topography**
Considering the rearrangement of enhancer accessibility in SGS NPCs, we decided to investigate the genome in the three-dimensional space where the contacts between distal and proximal gene regulatory regions happen and may even be revealed[51]. We, therefore, performed chromosome conformation capture by in situ Hi-C[52,53] in triplicate using one SGS NPC line (D868N) and its isogenic control line (D868D). Hi-C experiments showed good quality metrics (Supplementary

Data 6) and were reproducible across replicates, and by comparing them with a previously generated dataset from the same cell type[54] (Supplementary Fig. 5a, b). Considering the high level of correlation, we generated unique contact matrices merging the three replicates, obtaining ~927 and ~912 million unique contacts for control and SGS cells respectively. These evidenced 6343 and 5369 contact domains of 210 Kb and 228 Kb mean lengths, which are consistent among them and with matrices obtained elsewhere[54] (Fig. 3a, Supplementary Fig. 5c). The 3D conformation in SGS NPCs appeared overall preserved, despite the frequency of contacts in mutant cells was slightly lower when compared to control cells (Fig. 3a–c).

Given that SET accumulation prevents physiological accessibility in distal regulatory regions enriched for CTCF binding sites (Fig. 2), we decided to investigate 3D chromatin loops. These loops are defined as point-to-point interactions that often coincide with enhancer-promoter pairs, put in contact by cohesin-mediated extrusion[51]. To test whether SET plays any role in physiological loop formation and/or maintenance, we analyzed the contact maps at high resolution (~1 kb)[53] and identified 14,168 loops in the control NPCs (Supplementary Fig. 5d, Supplementary Data 7). Overall, out of 14,168 called loops, 21% (2978 loops) were weakened by at least 1.5-fold in SGS NPCs (Fig. 3d, Supplementary Data 8). Aggregate peak analysis (APA) confirmed that contact frequencies were decreased in this subset of loops (Supplementary Fig. 5e), that were longer than those unchanged (Supplementary Fig. 5f). Interestingly, the anchors of SGS-vulnerable loops displayed loss of both chromatin accessibility and H3K27 acetylation, particularly evident in anchors distal to the transcription start sites (TSSs) (Fig. 3e). The transcription levels of the genes close (within 10 Kb) to the anchors of affected loops were deregulated between control and SGS cells[39] (Fig. 3f, g). Gene ontology (GO) analysis demonstrated that these genes are enriched for biological processes important for developmental processes in general and for neural differentiation specifically (Fig. 3h, Supplementary Data 8). For instance, the loop that connects the promoter of the *RSPH3* gene (involved in the regulation of the cilium)[55] with the distal regulatory region within the *SYTL3* gene was weaker in SGS NPCs, due to the decrease in acetylation and accessibility of the distal anchor while *RSPH3* promoter was unaffected (Fig. 3i). A similar pattern was observed in the *RIMS4* locus (synaptic protein connected with autism cases)[56]. Notably, both genes resulted reduced in mutant cells in relation to the controls (Supplementary Fig. 5g).

In summary, our analyses indicate that in SGS condition, the chromatin looping is weakened with lesser H3K27 acetylation and chromatin accessibility, ultimately leading to aberrant expression of the associated genes.

**Neuronal development is regulated in an alternative way in SGS**
Neuronal specification from neural progenitors requires an extensive transcriptional remodeling coupled with epigenetic reorganization[57]. To assess whether this process is impaired in SGS, we longitudinally

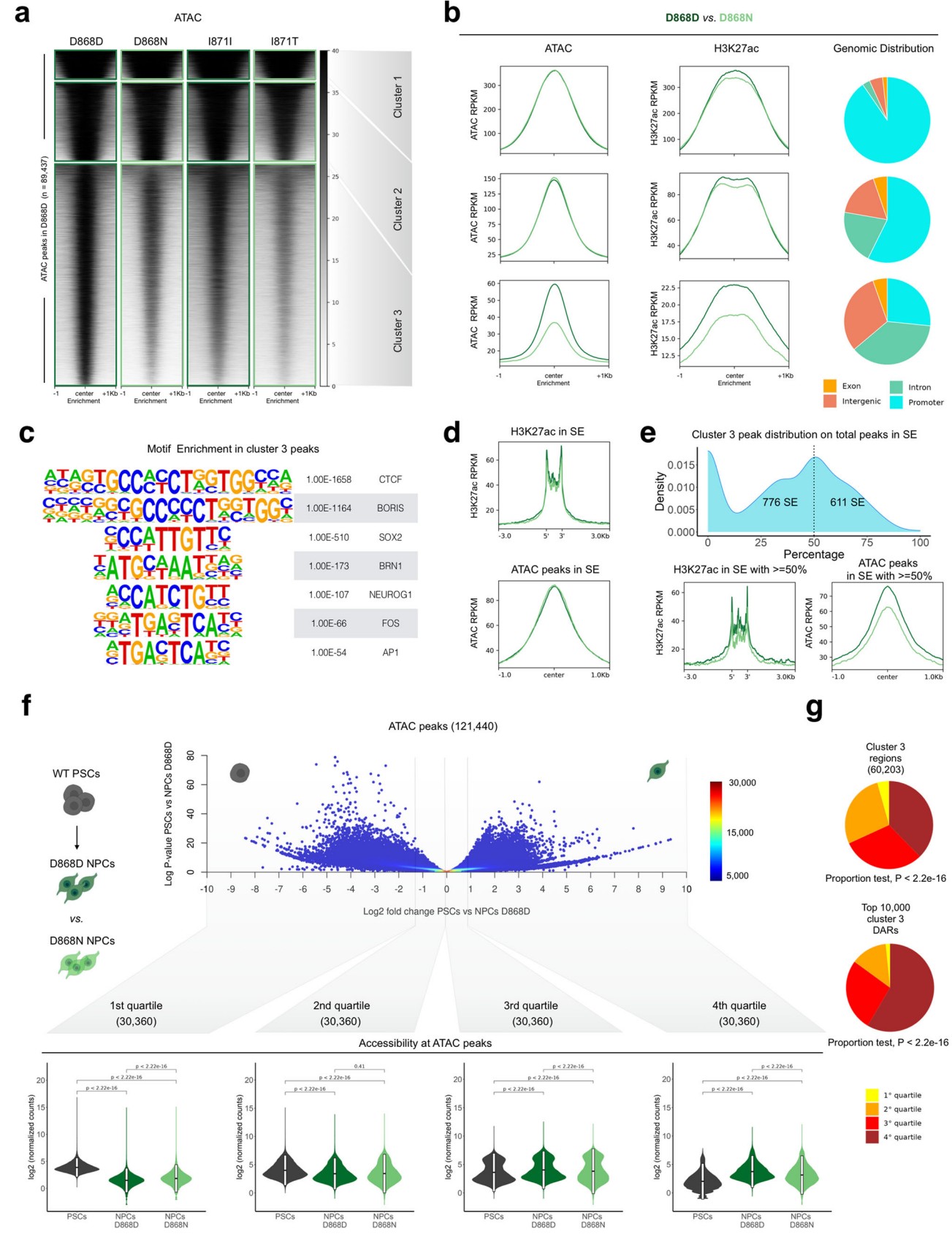

compared the dynamics of DNA accessibility using ATAC-seq. To do this, we performed ATAC-seq also in NPC-derived neurons, which confirmed the loss of accessibility in distal regions already seen in NPCs (Supplementary Fig. 6a, Supplementary Data 9). We previously reported that SGS mutant neurons can inherit cellular impairment (e.g., DNA damages), possibly unrelated to SET function on chromatin but able to interfere with, only when derived from NPC intermediate state[39]. Therefore, we repeated the ATAC analysis in neurons derived

**Fig. 2 | SET accumulation impairs chromatin accessibility at distal regulatory region. a** Heatmap of ATAC normalized signal in control (D868D) NPC peaks, $n = 89,437$. Regions are clustered using k-means ($n = 3$), cluster1 = 8,033, cluster2 = 21,201, cluster3 = 60,203. **b** Density plot of normalized ATAC-seq and H3K27ac ChIP-seq signal in regions inside clusters of a, the median value of each sample inside all regions analyzed was plotted. Pie charts represent the genomic distribution of regions inside clusters of (**a**). **c** Homer motifs enrichment in cluster3 result summary. **d** Density plot of normalized ATAC-seq and H3K27ac ChIP-seq signal in super-enhancers (SEs), $n = 1387$. The median value of each sample inside all regions analyzed was plotted. **e** Top, density plot of SE based on the percentage of cluster 3 ATAC peaks present inside each SE. Bottom, density plots showing H3K27ac ChIP-seq (left) and ATAC-seq (right) normalized signal inside SEs with more than 50% of associated ATAC peaks belonging to cluster 3 of (**a**), $n = 611$. **f** Summary results of open chromatin dynamic analysis from PSCs to NPCs. Top, adapted volcano plot shows how open chromatin regions ($n = 121,440$) are ordered based on log2 fold change between PSCs and NPCs and the cutoff value of the four quartiles (1st: −10 to −1.35; 2nd: −1.34 to −0.027; 3rd: −0.026 to 0.9; 4th: 0.9 to 10). Color legend represent the number of regions summarize by each data point. Bottom, violin plots of ATAC-seq normalized counts are plotted for the associated regions each quartile ($n = 30,360$, statistic two-sided Wilcoxon-test. Boxplot with 25–75th percentiles, mean, and whiskers of minima to maxima). Drawings created with BioRender.com. **g** Pie charts showing the proportion of all (top) and the top 10,000 differential (bottom) ATAC peaks contained in cluster 3 in (**a**) found in each of the four quartiles.

directly from iPSCs[58]. Indeed, SGS iPSCs, which showed neither SETBP1 nor SET accumulation, differentiate into neurons avoiding NPC-related confounding factors[39]. Importantly, we confirmed the SET-associated chromatin phenotype in SGS neurons derived from direct differentiation (Supplementary Fig. 6b, Supplementary Data 9). A straight comparison of open chromatin peaks between control NPCs and their derived neurons indicated differential chromatin usage either by gaining (14,822 peaks) or losing accessibility (12,486) during neuronal differentiation in vitro (Fig. 4a). The same approach in mutant NPCs to neurons transition revealed a comparable number of modulated regions (13,176 gain and 9937 loss) (Fig. 4a). Anyway, at the qualitative level, the regions between the two genotypes were different, with only 4488 and 2796 peaks that gained and lost accessibility, respectively, in common between control and mutant condition during the NPC-to-neuron differentiation (Fig. 4b, Supplementary Fig. 6c, Supplementary Data 9). Next, we enquired whether genotype-specific TF-driven regulatory dynamics are present within the regions that gain accessibility during neuronal specification. Using chromVAR[59] to estimate TF binding site (TFBS) accessibility, we found differential TFBS opening between control and SGS cells (Supplementary Fig. 6d, Supplementary Data 9). We also used chromatin deviations within TFBS to interrogate for cell- and genotype-specificity, obtaining a well separated clustering between the samples through the t-distributed stochastic neighbor embedding (tSNE) method[60] (Fig. 4c). Interestingly, a group of TFBS that normally increase their accessibility from NPCs to neurons, e.g., the sites recognized by the NR2F1 differentiating factor, had a different pattern in SGS, being already accessible in their parental NPCs and often closed in differentiated neurons (Fig. 4c, Supplementary Fig. 6d). We also noted that SGS neurons displayed excessive accessibility in other TFBS, often due to failure in closing NPC specific TFBS, e.g., EMX1 sites (Fig. 4c, Supplementary Fig. 6d). Integration with RNA-seq expression data did not pinpoint to a clear dysregulation of the corresponding TFs, which implies that the different TFBS accessibility were not induced by overexpression/repression of the corresponding factors in trans (Supplementary Fig. 6d).

Then, we wondered whether the different usage of DNA regulatory regions was affecting the dynamic of chromatin loops during neural differentiation. To do this, we performed Hi-C on NPC-derived neurons of control and SGS lines, as done for NPCs, obtaining comparable contact matrices, and again performing a comparison with a similar dataset[54] (Supplementary Fig. 6e–i, Supplementary Data 10). Longitudinal comparison of chromatin loops from NPC to the neuronal stage in the control line indicated 28% and 16% of the loops that increase and lose strength during differentiation, respectively (Fig. 4d, Supplementary Data 11). On the other hand, we observed 33% of stronger loops and 7% that were weaker in SGS neurons although a higher number of total loops identified (Fig. 4d, Supplementary Data 11). In agreement with the ATAC-seq data, the comparison of the anchors of the neuronal-specific loops (i.e., the loops that significantly gain strength in neurons compared to NPCs) in the two genotypes indicated only a small subset of loop anchors in common (Fig. 4e, Supplementary Data 11).

Next, we assessed whether the identified differential chromatin dynamic impacts gene regulation during neuronal differentiation. To take into account accessibility data and 3D genome architecture, we mapped each peak that gain accessibility in neurons (Fig. 4a, b) within the contact frequency maps coming from the Hi-C, to identify the most likely contacting gene (see Methods). Despite the large difference in the peaks between control and SGS (Fig. 4b), we found that most of the identified genes were in common (Fig. 4f, Supplementary Data 12) suggesting that, instead, a differential chromatin regulation on the same genes is in place during the neuronal differentiation in the two genotypes (Fig. 4g). These commonly regulated genes are enriched for GO term related to neuronal differentiation while very few of those that are either control- or SGS-specifically regulated appeared important for this process (Fig. 4h, Supplementary Data 12). RNA-seq analysis demonstrated a high level of transcriptional alteration of this group of genes between control and SGS neurons (Fig. 4i, Supplementary Data 13). An example is provided by the *SEMA3A* gene (encoding for the Semaphorin-3A, a secreted factor involved in neurite growth and neuronal migration)[61], whose promoter displayed differential promoter-enhancer contacts that were either preserved (e.g., #1 in Fig. 4j), gained (e.g., #2 in Fig. 4j), or lost (e.g., #3 in Fig. 4j) in mutant cells, globally reducing *SEMA3A* expression (Fig. 4j).

Altogether, our data describe a reorganization of chromatin regulation of gene transcription during neuronal differentiation in which the high level of SET hinders the accessibility, and thus the usage, of many putative physiological enhancers, while promoting the employment of an alternative set of distal regulatory regions as an adaptive mechanism in SGS.

## HDAC inhibitor fosters the maturation of SGS neurons
The data we have collected so far may explain that NPC-derived SGS neurons appear immature (i.e., simpler dendritic tree compared to controls) as we previously reported[39]. However, the DNA damage accumulation, the strong activation of PARP-1 signaling, and the early degenerative processes that we observed in these neurons[39] may be at least a concurring cause. Thus, we first checked the neurons derived directly from iPSC, that are free from NPC-derived impairment[39] and retain chromatin defects (Supplementary Fig. 6b, Supplementary Data 9). In line with our previous findings, mutant neurons presented shorter extensions and less arborization by Sholl analysis (Fig. 5a). The immature phenotype, possibly resulting from a blocked or delayed differentiation, was further confirmed by electrophysiological analysis. Whole cell patch clamp recordings from mutant neurons showed defective firing and reduced synaptic activity compared to control (Fig. 5b). Also, passive cell properties appear to be affected in mutants (Fig. 5b).

Following the idea that chromatin defects may be due to the cause of this phenotype, we tested whether interfering with the loss of histone acetylation can boost the maturation of SGS neurons. We employed the histone deacetylase (HDAC) inhibitor suberoylanilide hydroxamic acid (SAHA)[62], which successfully restored the H3

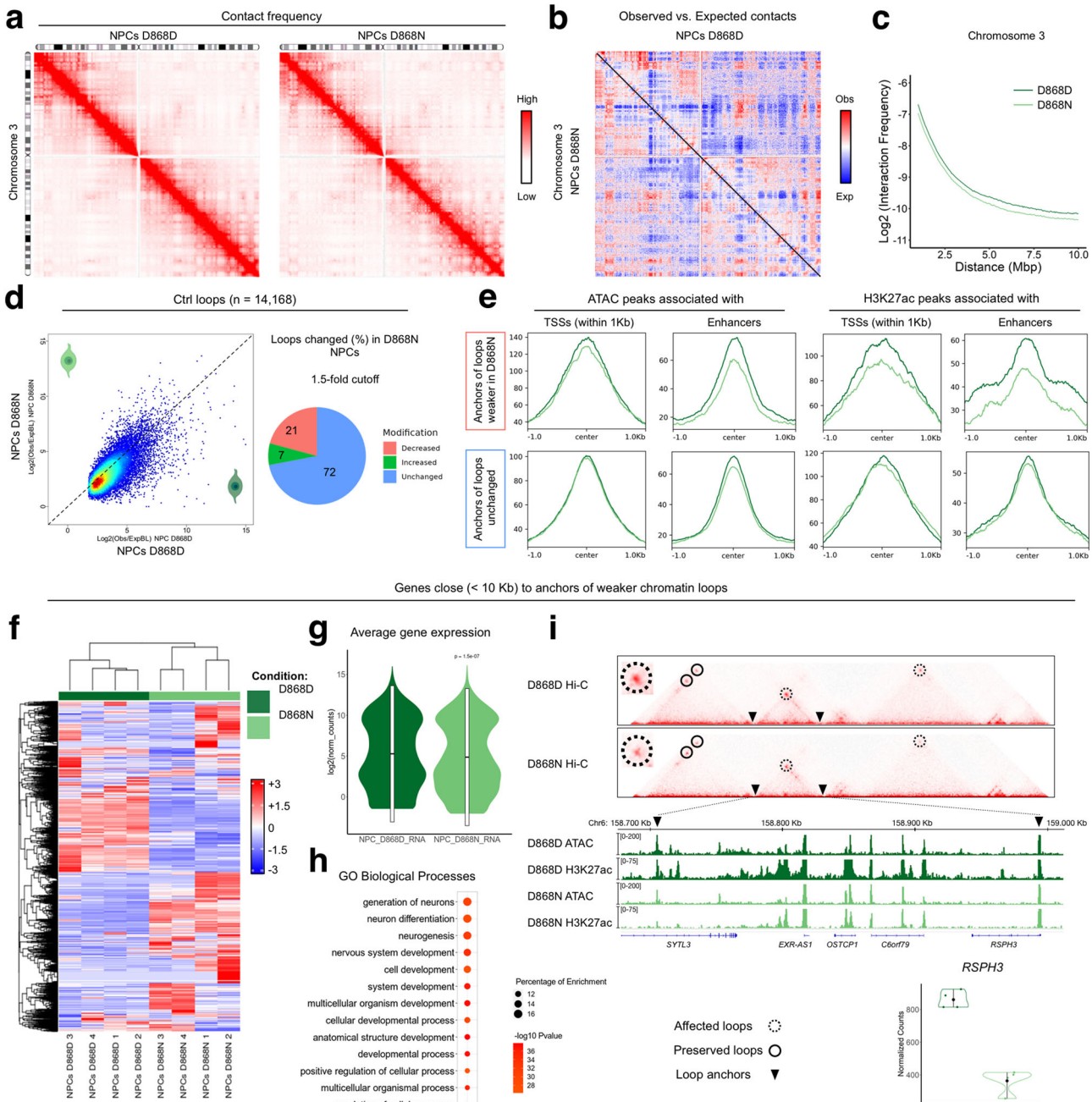

**Fig. 3 | Chromatin loops contact are altered in SGS NPCs. a** Normalized chromosome 3 interaction frequency maps of control (D868D) and mutant (D868N) NPCs. **b** Normalized chromosome 3 observed/expected interaction frequency map comparing control (D868D) and mutant (D868N) NPCs. **c** Plot showing the relation between the interaction frequency value and the contact distance between bins at 50 kb resolution comparing control (D868D) and mutant (D868N) NPCs. **d** Left, scatterplot representing the results of loop strength calculation of control (D868D) vs mutant (D868N) NPCs of the control (D868D) NPCs loop list (n = 14,168). Right, piechart shows the percentage of loops that exhibits an increase or a decrease over a 1.5-fold cut-off. **e** Density plots of ATAC-seq and H3K27ac ChIP-seq normalized signal in the respective significant peaks inside loop anchors belonging to loops with either weaker (top, n = 2978) or unchanged (bottom, n = 11,190) strength. Median value of each sample inside all regions analyzed was plotted. ATAC peaks in TSS +/− 1 kb, decreased n = 1478, unchanged n = 9401; ATAC peaks in enhancers,

decreased n = 931, unchanged = 3815; H3K27ac peaks in TSS +/− 1 kb, decreased n = 1482, unchanged n = 6990; H3K27ac peaks in enhancers, decreased n = 648, unchanged = 2747. **f** Heatmap showing expression levels of genes close (<10 kb) to loop anchors showing decrease strength (n = 5681). **g** Violin plot of the average normalized counts of the gene close (<10 kb) to anchors of loops showing decrease strength (n = 5681, statistic two-sided Wilcoxon-test, P = 1.5e-07. Boxplot with 25–75th percentiles, mean, and whiskers of minima to maxima). **h** Functional enrichment result on genes close (<10 kb) to loop anchors showing decrease strength (n = 5681). **i** Genomic locus of *RSPH3* gene showing interaction frequencies maps with underlined chromatin loops and genome browser of ATAC-seq and H3K27ac ChIP-seq of control (D868D) and mutant (D868N) NPCs. Mean RNA expression level of *RSPH3* in control (D868D) and mutant (D868N) NPCs is shown (bottom-right).

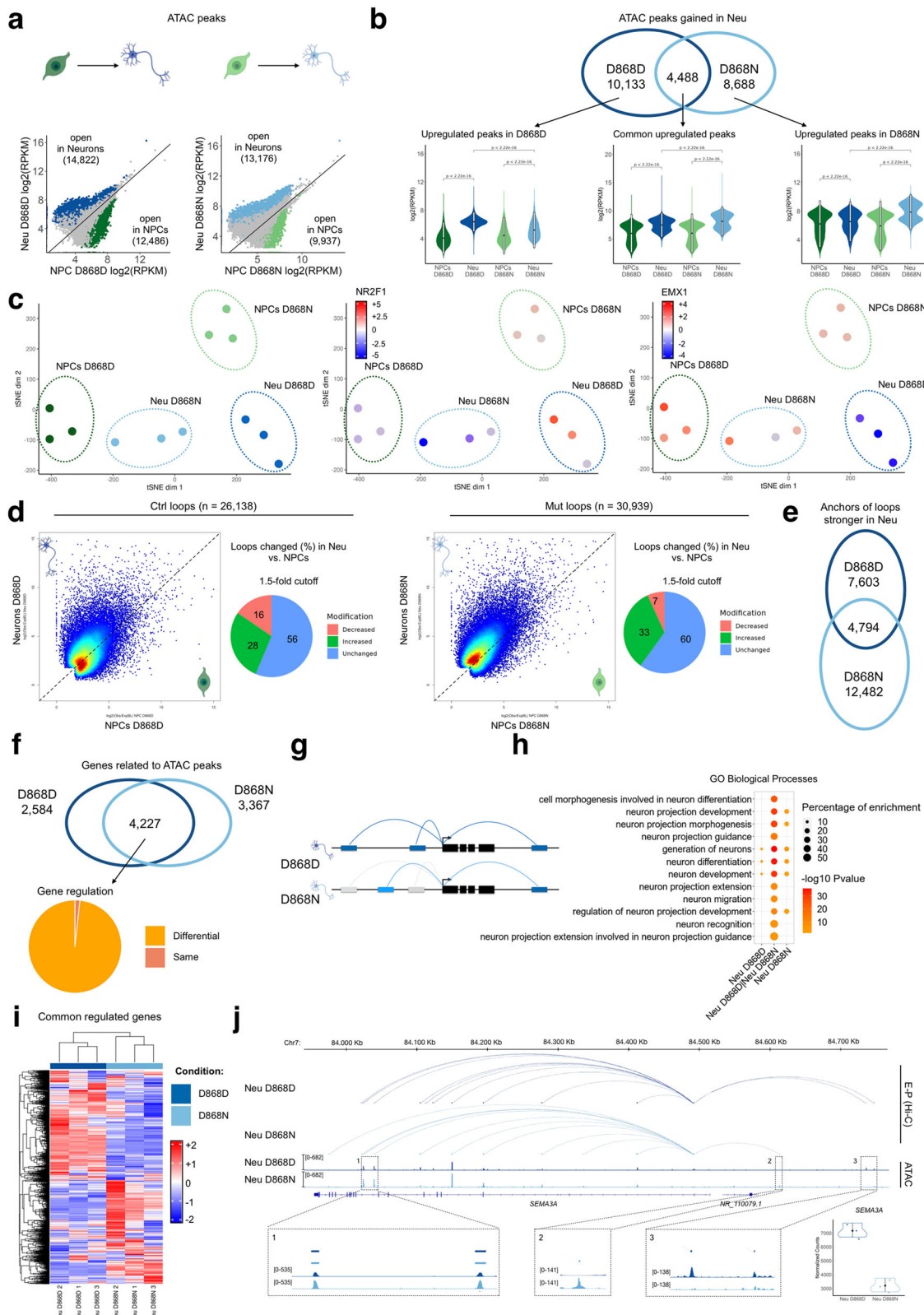

acetylation levels in SGS NPCs (Supplementary Fig. 7a). Interestingly, SAHA treatment could ameliorate the neuronal shape of the mutant neurons, likely allowing the correct expression of genes impacted by the high level of SET (Fig. 5c and Supplementary Fig. 7b). Conversely, the usage of the small molecule Nutlin-3a an inhibitor of P53 degradation that protects SGS NPCs against genotoxic damage[39] was shown ineffective (Fig. 5c and Supplementary Fig. 7b).

These results are in line with the model by which chromatin acetylation defects, secondary to high levels of SET, are responsible for defective neuronal differentiation in SGS.

## Chromatin rewiring in SGS in vivo model

To strengthen our conclusion using a more physiological model, we generated the first SGS murine model so far. We crossed the recently

**Fig. 4 | SGS neurons displays an altered gene regulatory architecture.**
**a** Correlation plots of normalized ATAC-seq signal between NPCs and NPC-derived neurons in the control (left, D868D) and mutant (right, D868N) condition. Each point represents a significant ATAC peaks in either condition ($n = 185,354$). Peaks with significant gain in accessibility are colored in blue (D868D, $n = 14,822$) or light blue (D868N, $n = 13,176$). Peaks with significant loss in accessibility are colored in green (D868D, $n = 12,486$) or light green (D868N, $n = 9937$). Drawings created with BioRender.com. **b** Top, Venn diagram showing overlap of peaks with significant gain in accessibility during neural development in control (blue, D868D, $n = 14,822$) and mutant (light blue, D868N, $n = 13,176$). Bottom, ATAC-seq normalized signal for each subset of peaks is represented in a violin plot (upregulated in D868D, $n = 10,133$, statistic two-sided Wilcoxon-test, $P < 2.22e$-16 for each comparison; commonly regulated, $n = 4488$, statistic two-sided Wilcoxon-test, $P < 2.22e$-16 for each comparison; upregulated in D868N, $n = 8688$, statistic two-sided Wilcoxon-test, $P < 2.22e$-16 for each comparison. All boxplot with 25–75th percentiles, mean, and whiskers of minima to maxima) **c** TFBS accessibility-based tSNE plot of ATAC samples of the D868 line, featuring regions that increase their accessibility during

neural differentiation in either genotype, $n = 23,309$. In the middle and right plots each sample is colored by the accessibility level of NR2F1 and EMX1 binding motifs found in the considered regions. **d** Scatterplot showing loop strength calculation in control loop list (left part, D868D. $n = 26,138$) and mutant (right part, D868N, $n = 30,939$). Pie chart shows the percentage of loops gaining or losing strength over a 1.5-fold cutoff. **e** Venn diagram showing the overlap of loops strengthened during development in control (D868D, $n = 14,766$) and mutant (D868N, $n = 20,566$). **f** Venn diagram showing the overlap of genes associated to ATAC-peaks significantly strengthened in neuronal development in control (D868D, $n = 6811$) and mutant (D868N, $n = 7594$). **g** Schematic model of differential gene regulation in the two conditions (D868D; D868N). **h** Functional enrichment result of shared genes in (**f**), ($n = 4227$). Statistic two-sided flase discovery rate (FDR). **i** Heatmap showing expression levels of shared genes in f ($n = 4227$). **j** Representation of the enhancer-promoter (E-P) regulatory network inside the *SEMA3A* gene locus (chr7: 83,918,959-84,834,917) showing E-P from Hi-C dataset and ATAC-seq tracks. Magnification in 1, 2 and 3 showing enhancers with differential accessibility between conditions. Mean RNA expression level of SEMA3A.

obtained *Rosa26-LoxP-STOP-LoxP-hSETBP1^G870S^* line (Crespiatico, Zaghi et al., submitted) (Fig. 6a), in which the human *SETBP1* carrying the SGS mutation G870S is silent, with Cre-expressing murine lines to induce specific SETBP1 accumulation. Since SGS is a systemic disease, we firstly employed the CMV::Cre line[63] to remove the STOP cassette and allow the overexpression of *SETBP1* in all the body's cells (Full mutant) (Supplementary Fig. 8a). While we failed to obtain live-born mutants, we were able to retrieve underdeveloped embryonic day 9.5–10.5 (E9.5–E10.5) mutant embryos retrieved (Supplementary Fig. 8a–c). Full mutant embryos, which expressed the GFP as proof of the correct expression of the transgene, displayed a reduced content of mature CD71⁺Ter119⁺ primitive erythrocytes within the yolk sac[64] (Supplementary Fig. 8c–e), which is likely to be the cause of the premature death. To circumvent embryonic lethality and having a particular interest in brain phenotypes, we decided to use the Nestin.:Cre murine line[65] to induce recombination in brain progenitors (Brain mutant) (Supplementary Fig. 8f). Brain mutants were born at Mendelian ratio and at Postnatal day 30 (P30) presented microcephaly, reduced brain structure (e.g., cortical wall) and increased ventricle volume (Fig. 6b, c, Supplementary Fig. 8g), as clinically reported for SGS patients[33]. By biochemical investigation on E14.5 embryos, we confirmed the expected high levels of both SETBP1 and its interactor SET (Fig. 6d). Interestingly, the decrease of H3K27 acetylation was also evident (Fig. 6d), sustaining (i) our in vitro data from iPSC-derivatives and (ii) the possibility of investigating the SGS SET-dependent chromatin remodeling in these animals.

To capture cellular heterogeneity in the cerebral cortex, we performed single-cell (sc) ATAC-seq and RNA-seq at the same time in the same individual cells using the Multiome approach (Chromium platform – 10x Genomics)[8]. We generated pooled libraries from three E14.5 and P2 cortices of both WT and Brain mutant animals (Supplementary Fig. 9a). Overall, we obtained 17,100 cells of control animals (5569 from E14.5 and 11,531 from P2) and 12,272 cells (1,238 from E14.5 and 11,034 from P2) from brain mutant mice after the quality check for both the techniques and filtering (Supplementary Data 14). To assess general similarities and divergences between single cells, we performed dimension reduction using uniform manifold approximation and projection (UMAP) analysis using RNA-seq dataset and identified and annotated clusters of the major cell types present during mouse cortical development[66], including apical radial glial cells (AP_RGC), intermediate neural progenitors (INP), excitatory neurons (ExN) from deeper (DL), upper layers (UL) and Layer 1 (L1), interneurons (IN), oligodendrocyte precursors (OPCs), astrocytes (Astro), Microglia, and endothelium, across the two stages and populated by both genotypes (Fig. 6e, Supplementary Fig. 9a, b). We also carried out dimension reduction through UMAP analysis based ATAC-seq data, maintaining the clustering and the same cell-type annotation as above. We

observed nice separation of the clusters (Fig. 6e, Supplementary Fig. 9a, b), confirming the expected concordance between transcriptomics and accessibility (Supplementary Fig. 9c) for cell-type specific analysis at least in brain[8]. RNA velocity analysis, which leverages the distinction between nascent and spliced RNA transcripts[67], was consistent with the identified clusters in both genotypes, also suggesting that the differentiating trajectories are conserved in SGS condition albeit slower in the timing in which they are executed compared to control (Fig. 6f).

To further study this aspect, we used ArchR to infer pseudotime from the ATAC-seq data[68]. By calculating the pseudotime trajectories along lineage specification of both DL and UL excitatory neurons, we confirmed that neuronal differentiation seemed delayed (Supplementary Fig. 9d). Later, we directly investigated the accessibility dynamics of the chromatin regions that featured the trajectories during the pseudotime. We discovered that i.e., for the development of DL neurons, the pattern of the physiological differentiation (control peaks) was altered in SGS animals that indeed employed a different set of regions (mutant peaks) (Fig. 6g). This difference is expected to impact on gene transcription. Accordingly, we found several hundreds of differentially regulated genes in each cluster, resulting in different levels of expression and/or different patterns of transcription during the developmental trajectories (Fig. 6h, Supplementary Fig. 9e, Supplementary Data 15). For instance, *Neurod6* (encoding for a neurogenic differentiation transcription factor)[69] and *Tbr1* (cortical early born neuron marker)[70] were downregulated in ExN_DL differentiation (Supplementary Fig. 9e), while *Nrxn1* (synaptic adhesion protein)[71] displayed a different model of activation during the pseudotime (Fig. 6h). To define whether a differential regulation of the same genes was in place during neuronal differentiation between the genotypes, we calculated at cluster level every single open chromatin region associated with each gene, as: "common" if found in both control and SGS cells and either "unique WT" or "unique MUT" if found only in control or mutant, respectively (Supplementary Data 16). The scatter plot featuring "unique WT" and "unique MUT" values indicated differential chromatin regulation (>2 in at least one of the two variables, meaning at least three chromatin regions gain/lost or both in SGS) in 7129 out of 17,445 genes (41%) in the ExN DL cluster (Fig. 6i), in line with what we have shown in vitro (Fig. 4). As an example, we show the differential regulation of *Nlgn1* gene (synaptic adhesion protein)[71] (Fig. 6j), that resulted less expressed in control than in SGS animals (Fig. 6i).

Next, we inspected the effect of the chromatin deregulation on the actual development of the organ, as one of the key clinical phenotypes of SGS. The excitatory neuron lineage showed a decreased distribution of SGS cells towards the end-point of the trajectory[72] ($P < 2.2 \times 10^{-16}$, two-sided Kolmogorov–Smirnov test; Fig. 6k), supporting delayed development of these neurons in the mutants.

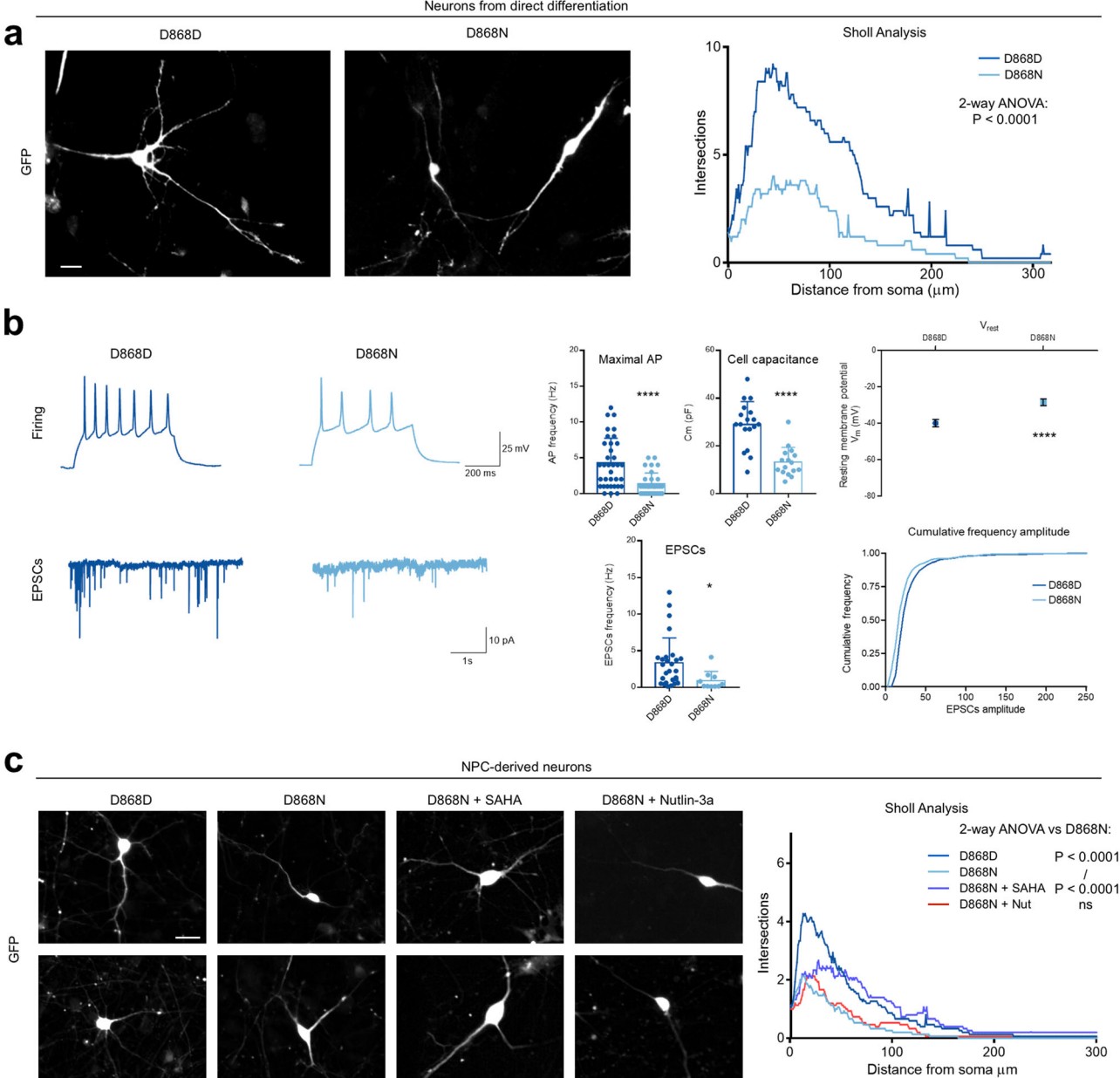

**Fig. 5 | SGS neurons phenotypic alterations are rescued by histone acetylation increase. a** Immunostaining for GFP on IPSCs derived neurons control (D868D) and mutant (D868N) and Sholl analysis result graphic (Statistic, 2-way-ANOVA, P = 0.0001). Bar = 10 μm. **b** Electrophysiological recordings of iPSC-derived control (D868D) and mutant (D868N) neurons. Example traces of firing capacity and EPSC current are shown (statistic two-sided *t*-test. Maximal action potential (AP), D868DvsD868N, ***P < 0.0001, n = 35 for D868D, n = 32 for D868N independent cells, from 3 independent experiments; Cell Capacitance, D868DvsD868N, ***P < 0.0001, n = 18 for D868D, n = 16 for D868N independent cells, from 3 independent experiments; Resting Membrane potential, D868DvsD868N, ***P < 0.0001,

n = 35 for D868D, n = 32 for D868N cells, from 3 independent experiments; EPSCs, D868DvsD868N, *P = 0.038, n = 27 for D868D, n = 10 for D868N independent cells, from 3 independent experiments; Cumulative frequency, n = 100 independent measurement. Data are presented as mean values +/− SEM). **c** Left, immunostaining for GFP on NPC-derived neurons both control (D868D), mutant (D868N), mutant treated with SAHA, mutant treated with Nutlin. Right, Sholl analysis result output (statistic 2-way-ANOVA, D868DvsD868N, P < 0.0001, D868DvsD868N SAHA, P < 0.0001, D868DvsD868N Nutlin, P < 0.0001, D868NvsD868N SAHA, P < 0.0001, D868NvsD868N Nutlin, P = ns, D868N SAHAvsD868N Nutlin, P < 0.0001). Bar = 20 μm.

Immunohistochemical examination of E14.5 embryos from both control and mutant littermates confirmed slow development in SGS, where apical RGCs (SOX2⁺) tended to accumulate at the expense of committed INPs (TBR2⁺) and early differentiating neurons (DCX⁺) (Fig. 6l), corroborating the cell fractions within clusters in the multi-ome experiment (Supplementary Fig. 9f). Finally, we performed in utero electroporation (IUE)[73] to address an expression plasmid encoding for a red fluorescent protein in cortical RGCs of control and mutant E13.5 embryos, to analyze neuronal output after that the

neurogenic processes have occurred. In accordance with the data obtained in iPSC-derived neurons (Fig. 4), neuronal shape in SGS mutants appeared simpler than in the WT counterparts (Fig. 6m).

Altogether, these results provide evidence of differential chromatin usage in the first in vivo model of SGS. This molecular rewiring is connected with altered level of gene expression as well as transcriptional activation dynamics along relevant developmental trajectories, e.g., from RGCs to excitatory neurons, that possibly lead to a delayed formation and maturation of the brain in SGS.

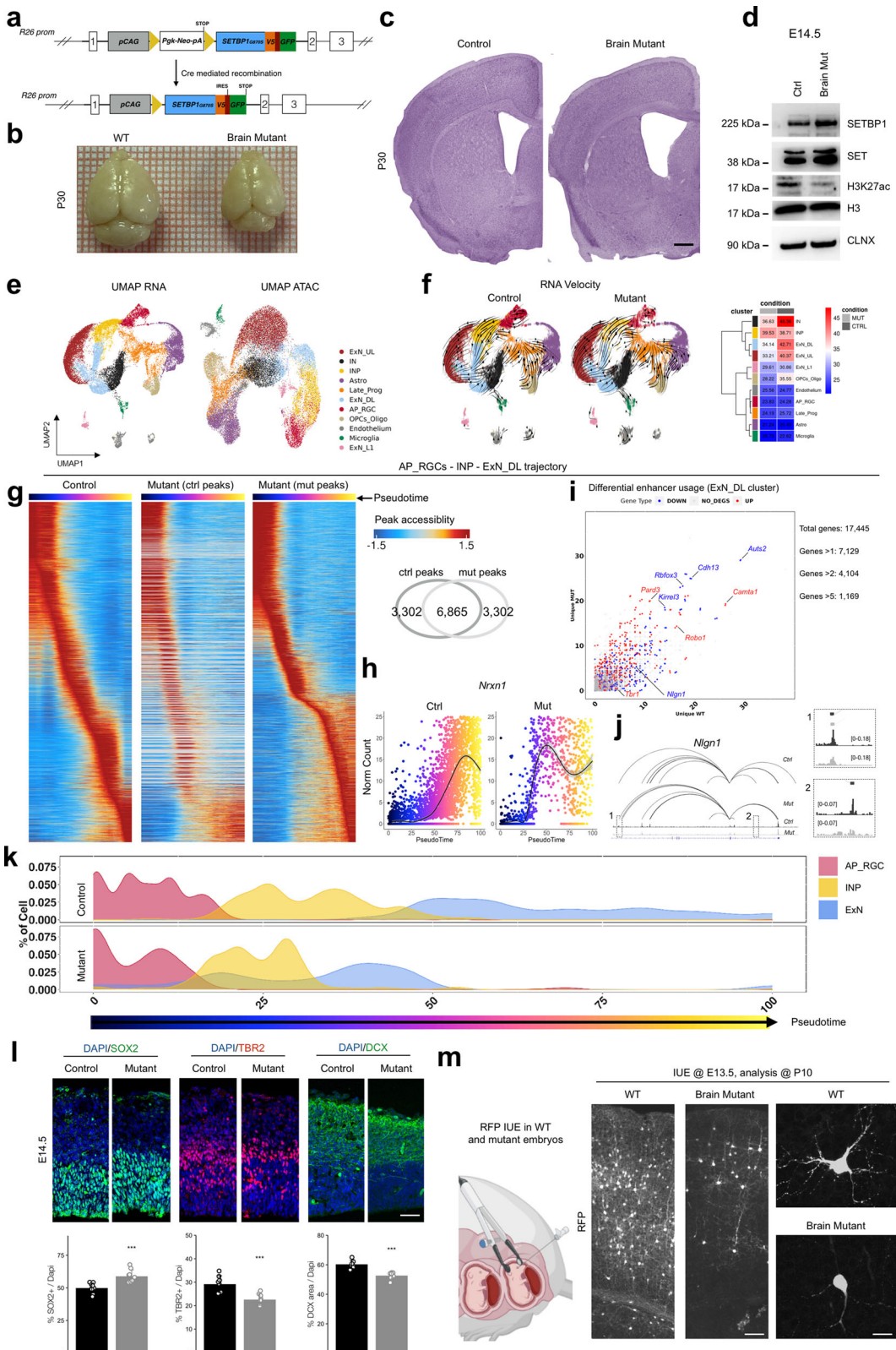

## Discussion

SET protein has been associated with histones and their modifications[23,26,34] as well as with severe human diseases, including cancers and SGS[27,28,34]. However, direct evidence of chromatin defects and their eventual connection with the pathogenesis of SET-associated diseases has never been investigated. Here, we combined bulk and single-cell transcriptomics, ChIP-sequencing, 3D genome architecture, zebrafish manipulation, mouse genetics, and electrophysiological recordings to address how high levels of SET hinder the usage of the physiological chromatin regulatory regions during cell fate acquisition.

Following the hypothesis that increased SET levels would lock stronger the histones and, thus, prevent acetylation of lysines in their tails, we generated both in vitro and in vivo models in which SET is

**Fig. 6 | SETBP1 overexpression induce brain developmental alterations in mouse model. a** Scheme of the modified *Rosa26* with mutant *SETBP1* (G870S) cDNA, before and after CRE recombination. **b** Brains of control (Cre-) and mutant (Cre+) P30 mice. Square side = 1 mm. **c** Nissl staining of P30 brains. Scale bar = 500 μm. **d** Western blot for SETBP1, SET, and H3K27ac levels: H3 and Calnexin (CLNX) are used as loading controls. **e** UMAP plots of scMultiome clusters. RNA (left) and ATAC (right) dataset. **f** UMAP of RNA velocity, each arrow shows the direction and speed of the velocity flow in both experimental conditions. Heatmaps on the left contains average velocity value for each cluster. **g** Heatmaps showing accessibility Z-score during AP_RGCs to ExN_DL trajectory. The left heatmap shows Z-score calculated in control condition, the middle heatmap shows Z-score calculated in control peaks using only mutant cells, while the right heatmap shows Z-score calculated in mutant condition. The Venn diagram shows overlap mutant and control peaks used in the analysis. **h** Scatter plot of *Nrxn1* expression along pseudo time trajectory of AP_RGCs to ExN_DL in control (left) and mutant (right).

**i** Differential gene regulation inside the ExN_DL cluster, each point represents a gene. Number of unique peaks associated to each gene, in each genotype, are plotted. Points in blue are downregulated genes, while those in red are upregulated. **j** Enhancer-promoter regulatory network inside the *Nlgn1* gene locus (chr3: 25,425,522-26,444,634), magnifications in 1 and 2 show differential accessibility at two enhancers ExN_DL cluster. **k** Pseudo time density plot of AP_RGC to ExN trajectory. **l** Immunostaining in E14.5 developing cortex. SOX2 marks apical radial glia, TBR2 intermediate progenitors and DCX marks early neurons (statistic two-sided *t*-test, SOX2, Control vs Mutant, ***P < 0.001, n = 9 independent embryos for each group; TBR2, Control vs Mutant, ***P < 0.001, n = 7 embryos for each group; DCX, ***P < 0.001, n = 7 independent embryos for each group. Data are presented as mean values +/− SEM. Bar = 40 μm. **m** Schematic representation of in utero electroporation (IUE) experiment in E13.5 embryos, created with BioRender.com. RFP immunostaining on IUE brains (P10) with magnification on the right. Left bar = 50 μm; right bar = 10 μm.

upregulated either by direct overexpression or as a consequence of *SETBP1* mutations. In all the models, we demonstrated that histone acetylation is impaired, confirming early pieces of evidence in cancer cells[74]. Unexpectedly, H3K27ac ChIPseq profiling, one of the major acetylated lysines in cells, showed an appreciably but not extensive decrease in genome-wide acetylation in SGS. This may be due to stoichiometric unbalance of SET-participating complexes, e.g., INHAT complex[23], that would limit SET functionality. However, since defects in histone acetylation are relevant to biochemistry, SET complexes are more likely working towards specific genomic loci than in a pervasive manner. We also showed that chromatin accessibility in SGS NPCs mirrored H3K27ac behavior. Interestingly, the most affected regions, i.e., those that lose acetylation and accessibility, are putative regulatory elements located far away from gene promoters where the levels of H3K27ac are generally sparse and presumably more sensitive to changes. An additional, not mutually exclusive possibility is that SETBP1 accumulation may cause direct effects per se in a SET-independent fashion, modifying the phenotype. We indeed demonstrated a direct SETBP1 binding on certain gene promoters as part of the HCF1/KMT2A/PHF8 epigenetic complex[75]. Thus, SETBP1 direct activation of gene promoters may either contribute as negative factor to the disease or participating in the attempt of rescue of the transcriptional landscape of SGS cells. Yet, the confirmation of enhancer-specific chromatin defects in SET-overexpressing models, iPSC, NPCs, and zebrafish embryos, in which SETBP1 is unaffected, advises for SET dependency.

Enhancers are thought to come into proximity to their target promoters through chromatin bending[51,76]. Studying 3D conformation of SGS chromatin, we found the weakening of a subset of chromatin loops in which the specific defect was at the level of the distal anchors rather than the promoters. Thus, the augmented SET levels induce the lack of proper acetylation in specific enhancers, the failure of their opening, and the subsequent loss of coupling with cognate promoters. The dynamic gain of a repertoire of newly acetylated and accessible chromatin sites is fundamental for the homeostasis of the correct developmental programs, from the pluripotency/differentiation dynamics, to fate establishment to the acquisition of specific functions[20,45,77,78] and is often hijacked as oncogenic mechanism[42,50,79]. We propose that a balanced amount of SET is key for epigenetic modification and functional usage of the correct array of regions. We found that a high percentage of the chromatin regions that are designated to be opened during the transition between cell stages at least of the neuronal lineage, i.e., from PSC to NPC and from NPC to neuron, show specific vulnerability in the SET^high condition. The SET binding preference for un-acetylated histones[26] that are not yet open but are poised to activation for developmental purposes, is in line with our findings. Alteration in this developmental-related chromatin dynamics by either reducing or increasing SET may lead to either counteracting or over-inducing differentiation programs. Notably, the

complete loss of *Set* compromises embryonic development in mice[80], and its partial inactivation leads to neurodevelopmental disorder[31], while SET accumulation is associated with cancer and SGS[27,28,34], all conditions that have been tightly connected with aberrant developmental programs. For instance, we reported a profound transcriptional dysregulation of SGS NPCs that resembles a mixture of different developmental programs[39]. Our whole body *SETBP1* overexpressing mouse dies early *in utero* for anemia. We believe that this is due to a strong ectopic expression of *SETBP1* along primitive hematopoiesis where/when it is normally off[81]. Recently, we also described that hematopoietic stem cells (HSCs), from *Rosa26-LoxP-STOP-LoxP-hSETBP1^G870S* crossed with Vav::Cre mice, transplanted in sub-lethally irradiated mice lead to lethal anemia (Crespiatico, Zaghi et al., submitted). These data confirm the importance of SETBP1 and likely SET in developmental processes of the blood, as already suggested by their mutations in cases of myeloproliferative diseases and leukemia[30,82]. However, in the context of in the blood diseases other functions of both proteins had been demonstrated to have a direct roles, including direct SETBP1 binding to DNA to foster gene transcription[75].

Neural development does happen in SGS, both in patients and in experimental models. We propose that other chromatin regions replace the physiological repertoire of enhancers to proceed with the differentiation. We found that SGS-specific putative enhancers increase their accessibility, contact the promoters, and eventually compensate for the regulation of the genes left orphans by the SET-sensitive regions. It has been proposed that the genes often display multiple enhancers with redundant or partially redundant spatio-temporal activity, called "shadow enhancers"[17]. The enhancers that insist on the same gene, may represent examples of shadow enhancers within a multi-enhancer architecture of that gene, at least in some cell type or developmental stage. Further investigation will elucidate this and other aspects, for instance how the activation of SGS-specific regions is escaping the inhibition by SET accumulation. The already mentioned capability of SETBP1 to work, in concert with other co-factors, as transcriptional activator[75] may be relevant in this matter. A further characterization of the direct role of SETBP1 in this context might elucidate the rewiring process occurring within the SGS chromatin.

Despite the attempt for adaptive compensation in SGS for neural specification to occur, we showed both in vitro and in vivo that the process remains inefficient, i.e., cortical progenitor numbers are unbalanced, neurons are immature and hypo-functional, and hundreds of relevant genes are dysregulated. The delayed neuronal maturation and the early degenerative processes, that we previously observed in NPC-derived neurons[39], may also concur explaining, at least in part, the severe brain malformations and neurological impairments of SGS patients[33].

In summary, we show that supra-physiological levels of the oncoprotein SET corrupt the normal usage of distal regulatory

elements that in turn ensure the fidelity of transcriptional programs during development and maturation. We provide evidence in the context of SGS, where SET is accumulated, neural fate specification and maturation are pursued, but not fully achieved, due to the usage of an alternative *in cis* regulation of the gene network used in the physiological process. This chromatin rewiring, on top of possible independent SETBP1-direct effects, causes transcriptional impairments thus providing an additional level of comprehension of the pathological basis of SGS. Future work will clarify whether this could be the case also for other SET-related diseases.

# Methods

## Mice

*Rosa26-LoxP-STOP-LoxP-hSETBP1$^{G870S}$* were generated by our group (Crespiatico, Zaghi et al., submitted). Nestin::Cre (strain #003771) mice and CMV::Cre (strain #006054) mice derived from Jackson Laboratory. All the mice strains were in C57Black6j background. Mice were maintained at the San Raffaele Scientific Institute Institutional mouse facility in a pathogen free environment. Temperature and air flow were controlled and constant as well as light/dark cycle. Experiments were performed in accordance with experimental protocols approved by local Institutional Animal Care and Use Committees (IACUC).

## Zebrafish maintenance

All animal procedures were performed in accordance with national (Italian decree March 4, 2014, n. 26) and European union (EU Directive 2010/63/EU) animal welfare guidelines. The fish were raised and maintained at 28 °C on a 14/10-h light/dark cycle at the San Raffaele Scientific Institute facility (Aut. Prot, 102093; Aut. Min, 06/2021-UT). Zebrafish embryos were obtained from natural matings of the Wild-type strain AB, raised in E3 medium at 28.5 °C and staged according to the reference guidelines. Before manipulations, embryos were dechorionated, washed and anesthetized with 0.016% tricaine tricaine (160 mg/l, Ethyl 3-aminobenzoate methanesulfonate salt; A5040, Sigma-Aldrich).

## TetOn-SET inducible cell iPSCs line generation

To obtain an inducible SET overexpressing iPSC line we infected NEOF wild type with two different lentiviruses, LV-TetON-SET-V5-PuroR and LV-Ef1a-tTa-BlastR. After infection, cells were selected for 3 days with both puromycin and blasticidin. Then, cells were growth in normal medium for 7–10 days and another round of antibiotic selection was performed for 3 additional days to obtain a pure cell line.

## SGS iPSCs lines

SGS iPSCs were the same used in our previous work[39], they are deposited on hPSC$^{reg}$ (https://hpscreg.eu) which released us certificates (#OSRi009-A, OSRi009-A-1, OSRi010-A, OSRi010-A-1). All iPSCs lines were cultured in feeder-free conditions in mTeSR1 (Stem Cell Technologies) supplemented with Pen/Strept (1%, Sigma-Aldrich) and seeded in six-well plates coated with human embryonic stem cell (HESC)-qualified Matrigel (Corning).

## Derivation and maintenance of NPCs

NPCs were derived using an optimized protocol previously published[83]. iPSCs were dissociated in cell clusters using Accutase solution (Sigma-Aldrich) and seeded onto low-adhesion plates in mTeSR1 supplemented with 0.5X N2 supplement (ThermoFisher Scientific), Pen/Strept (1%, Sigma-Aldrich), human Noggin (0.5 μg/ml, R&D System), SB431542 (5 μM, Sigma-Aldrich) and ROCK inhibitor Y27632 (10 μM). Medium was replaced every 3 days. After 10 days, embryoid bodies were seeded onto matrigel-coated plates (1:100, Matrigel growth factor reduced, Corning) in DMEM/F12 (SigmaAldrich) supplemented with 1X N2 supplement, non-essential amino acids (1%, MEM NEAA, ThermoFisher Scientific) and 1% Pen/Strept. Medium was

replaced every other day. After 10 days, rosettes were dissociated with Accutase solution and NPCs (P0) were plated onto matrigel-coated flasks in NPC media containing DMEM/F12, 0.5X N2, 0.5X B27 supplement (ThermoFisher Scientific), Pen/Strept (1%) and bFGF (20 ng/ml, ThermoFisher Scientific). NPCs were passaged twice a week using Accutase solution; experiments were performed with NPCs between P3 and P10.

## Neuronal differentiation from NPCs

NPCs were derived as previously described[39]. Briefly, at day 0 medium was replaced with differentiation medium composed by NPC medium without bFGF, supplemented with SU5402 (Sigma-Aldrich, 10 μM), PD0325901 (Sigma-Aldrich, 8 μM) and DAPT (Sigma-Aldrich, 10 μM). Differentiation medium was replaced every day with fresh one at day 1 and 2. At day 3 cells were detached by Accutase solution and incubated at 37 °C for 20 min to obtain a single cell suspension.

Cells were centrifuged (0.5 g, 5'), counted and seeded at a density of 55,000 cells/cm² onto PolyL-Lysine/Laminin/Fibronectin (Sigma-Aldrich, 100 μg/ml, 2 μg/ml, 2 μg/ml) coated plates or coverslip in neuronal maturation medium supplemented with ROCK inhibitor Y27632 (10uM) for the first 24 h. Neuronal maturation medium was composed by Neurobasal A (ThermoFisher Scientific) supplemented with 1X B-27 supplement, 2 mM Glutamine, 1% Pen/Strept, BDNF (Peprotech, 20 ng/ml), Ascorbic acid (Sigma-Aldrich, 100 nM), Laminin (Sigma-Aldrich, 1 μg/μl), DAPT (Sigma-Aldrich, 10 μM), dbcAMP (Selleckchem, 250 μM). Culture medium was replaced the next day to remove ROCK inhibitor, and then half of the medium was replaced with fresh Neuronal maturation medium twice a week.

## Direct neuronal differentiation from iPSCs

At the start of differentiation protocol cells were infected (at differentiation day −2) using the LVTetOn-Neurog2-T2A-PuroR and LV-Ef1a-tTa vectors[58] in mTeSR1 medium supplemented with Doxycycline (2 μg/ml), overnight. The following day, cell medium was changed with fresh mTeSR1 medium supplemented with antibiotic selection (Puromycin 1 μg/ml) and Doxycycline; Doxycycline was kept for all the experiment. At day 0 medium was exchanged with differentiation medium "mTeSR1 + LSBX". Differentiation medium was replaced daily according to the following scheme:

    Day 0,1: mTeSR1 + LSBX
    Day 2,3: mTeSR1 + LSBX + PSD
    Day 4,5: 2/3 mTeSR1 + 1/3 N2 medium + LSX + PSD
    Day 6,7: 1/3 mTeSR1 + 2/3 N2 medium + PSD

At day 6 cells were detached by Accutase solution (Sigma-Aldrich) and incubated at 37 °C for 20 min to obtain a single cell suspension. Cells were centrifuged, counted and seeded at a density of 55000 cells/cm² onto Poly-L-Lysine/Laminin/Fibronectin (all from Sigma-Aldrich, 100 μg/ml, 2 μg/ml, 2 μg/ml) coated plates or coverslip in neuronal maturation medium supplemented with ROCK inhibitor Y27632 (10 uM) for the first 24 h. Neuronal maturation medium was the same used for NPC-derived neuronal cultures. Cells were routinely cultured in neuronal maturation medium for the duration of the rest of the experimental protocol.

LSBX: LDN193189 (Stemgent, 250 nm), SB431542 (Sigma-Aldrich, 10 μM) XAV939 (SigmaAldrich, 5 μM). PSD: PD0325901 (8 μM), SU5402 (10 μM), DAPT (10 μM). N2 medium: DMEM/F12 with B27 supplement (0.5X) and N2 supplement (0.5X).

## Cell treatment with Suberoylanilide hydroxamic acid (SAHA)

NPCs cells were plated into Matrigel-coated 10 mm² plate and treated for 3 days with SAHA (20 nM, Sigma-Aldrich, in DMSO (0.05%, Sigma-Aldrich) in culture media before performing western blot analysis. The addition of the same volume of DMSO was used as control.

Neurons were treated with either SAHA (20 nM) or Nutlin-3a (Sigma-Aldrich) or DMSO from day 0 in culture media.

## Immunofluorescence on cultured cells

Cells were plated on coated glass coverslips (13 mm) and were fixed for 20 min on ice in 4% paraformaldehyde (PFA, Sigma), solution in phosphate-buffered saline (PBS, Euroclone). Two washes were performed afterward with PBS, cells were then permeabilized for 30 min in blocking solution, containing 0.2% Triton X-100 (Sigma-Aldrich) and 5% donkey serum (SigmaAldrich), and incubated overnight at 4 °C with the primary antibodies diluted in blocking solution. The next day, cells were washed 3 times with PBS for 5 min and incubated for 1 h at room temperature with Hoechst 33342 (ThermoFischer Scientific) and specific secondary antibodies (ThermoFisher Scientific) in blocking solution. Images were acquired with epifluorescence microscope Nikon DS-Qi2 and analyzed with ImageJ software.

## Immunofluorescence analysis and imaging on E9.5 embryos

E9.5 yolk sacs (YS) were extracted from embryos and fixed for 2 h at 4 °C in 4% paraformaldehyde[84,85]. After antibody labeling, YS were placed in a solution of 50% glycerol in PBS at 4 °C overnight, and subsequently flat-mounted. Image acquisition was carried out at room temperature with a Zeiss LSM-710 confocal system with an EC Plan-Neofluar 40x/1.30 Oil DIC M27 objective. Image processing was carried out using Zeiss Zen lite software and Adobe Photoshop CC 2019. Bright field images of whole embryos and YS were taken with a Leica EZ4D stereomicroscope.

## Immunofluorescence on embryonic and post-natal day 10 mouse brain

Brains of E14.5 embryos were fixed in 4% paraformaldehyde (PFA, Sigma-Aldrich) overnight at 4 °C, post dissection. P10 brains were extracted from the skull after brief perfusion with Sodium Chloride 0,9% (S.A.L.F) and PFA 4% in PBS (PFA, Sigma-Aldrich), a final fixation step overnight at 4 °C again in PFA 4%. The day after, 3 washes in PBS were performed. Overnight precipitation in 30% sucrose in water was executed right-after. Post precipitation brains were frozen in OCT (VWR) embedding medium. Brains were froze-cut in a cryostat at a 10 μm thickness and collected on pretreated glasses.

For immunofluorescence, brain sections were boiled for 5 min in citrate buffer when required depending on the primary antibody. After 2 washes in PBS for 5′, permeabilization was performed for 1 h in blocking solution containing 0.2% Triton X-100 (Sigma-Aldrich) and 10% donkey serum (Sigma-Aldrich), subsequent incubation was performed overnight at 4 °C with the appropriate primary antibody in blocking solution. The next day, 3 washes of 5 min each were performed in PBS, sections were then incubated with appropriate fluorofore-conjugated secondary antibody diluted in blocking solution for 1 h at room temperature. After incubation slices were washed again 3 times for 5 min in PBS, then were mounted using fluorescence mounting media (Dako) with a glass coverslip. Images were acquired using either TC SP5 or TC SP8 confocal microscope (Leica), and analyzed with ImageJ software. A list of antibodies is provided in the Supplementary Table 1.

## Nissl staining on adult mouse brain

Mice were anesthetized with $CO_2$ and perfused through the heart, first with Sodium Chloride 0.9% (S.A.L.F) to remove blood form the tissues and then fixed with Paraformaldehyde 4% in PBS (PFA, Sigma-Aldrich). Brains were extracted from the skulls and left in PFA 4% overnight. To remove water from the tissues, brains were dipped in Sucrose 30% until they sank in the bottom of the falcon. Subsequently, they were included in OCT compound, frozen and finally cut with the cryostat in 50 μm coronal section, slices were conserved in PBS-azide 0.1%.

To perform Nissl staining, brain sections are laid in gelatinized slides and once dried, slices are briefly washed in $H_2O$ mQ and then stained with 0.1% Cresyl violet solution, previously heated at 65 °C, for 8 min. After two washes of 3 min in $H_2O$ mQ, slides are dehydrated with ethanol scale, dipping the slides in ethanol 70%, 95%, and 100% for 2 min, and lastly in xylene for 30 min.

The mounting medium used is Eukitt®.

## Flow cytometry

Single-cell suspensions were prepared from *Rosa26::LoxP-STOP-LoxP-hSETBP1*[G870S/+] (WT) and *Rosa26::hSETBP1*[G870S/+] (full mutant) embryos YS. After collagenase digestion, cells were resuspended in Calcium Magnesium-Free PBS, FBS 10%, Penicillin-Streptomycin 1%, EDTA 2 mM and stained with Rat anti-mouse CD71-PE (BioLegend Cat# 113808, RRID:AB_313569), Rat anti-mouse Ter119 -PE-Cy5 (BioLegend Cat# 116210, RRID:AB_313711). Gates were set using unstained, single stained and fluorescence-minus-one (FMO) controls. Dead cells were excluded based on Hoechst 33258 (Sigma) incorporation. Data acquisition was performed on a BD LSR Fortessa X-20 cytometer. Flow cytometry data was analyzed using FlowJo software v10.8.1 (BD).

## Generation of mRNA and Zebrafish embryo microinjection

Polyadenylated mRNAs were produced using the mMessage mMachine SP6 Transcription Kit (AM1340, ThermoFisher Scientific) and Poly(A) Tailing Kit (AM1350, Ambion). The resulting mRNAs were purified with QIAGEN RNeasy mini kit (74104) and quantified by nanodrop spectrophotometer. RNA quality was also visualized on agarose gel. 2.5 nl of mRNA were injected into the yolk of one cell stage zebrafish embryos at a concentration of 100 or 200 ng/ul. The injections were performed with an air-pressured Pico-lighter injector (PLI100A, Warner Instruments) using 0.5 mm glass capillaries (30-0032, Harvard apparatus) pulled with a PC-10 needle puller (Narishige).

## Western blotting

iPSCs, NPCs, neurons, mouse cortex or Zebrafish brains were homogenized in RIPA buffer (50 mM Tris pH 7.5, 150 mM NaCl, 1 mM EDTA, SDS (0,1% for cells, 1% for cortex)), 1% Triton X-100, Roche Complete EDTA-free Protease Inhibitor Cocktail, Roche PhosSTOP EASYpack) and Western blot analysis was performed using primary antibodies as needed, incubated overnight at 4 °C in blocking solution composed by 5% BSA (Sigma-Aldrich) or 5% Non-Fat Dry Milk in PBS-TWEEN 0,1% (Sigma-Aldrich). Band densitometry relative to control was calculated using Image Lab software (Biorad), normalized on housekeeping as indicated in each figure; post-translational modifications were analyzed by first normalizing both total and modified form of the protein of interest on housekeeping, and then plotting the ratio between modified vs total relative to control. A list of antibodies is provided in the Supplementary Table 1. Cropped images are shown in the figure panels while original uncropped images are provided in Supplementary Figs. 10 and 11.

## Immunoprecipitation

NPCs were lysed in IP buffer (50 mM Tris pH 7.4, 150 mM NaCl, 1 mM EDTA,1 mM EGTA, 1% Triton X-100). After preclearing with magnetic beads (Protein G Dynabeads, Life Technologies), 2 h at 4 °C on a rotating wheel. Before antibody incubation, a portion of each sample was saved as input. Samples were incubated overnight with the required antibody on a rotating wheel. The next day IP was performed using Protein G coated magnetic beads for 2 h at 4°. After 3 washes, samples were boiled at 70°. WB was performed to reveal the result.

## Electrophysiology

Patch clamp recording and analysis. Electrophysiological characterization of iPSC-derived neurons was carried out adopting patch clamp in whole cell configuration. Cells were perfused with ACSF containing (in mM: 125 NaCl, 25 $NaHCO_3$, 2.5 KCl, 1.25 $NaH_2PO_4$, 11 Glucose, 2 $CaCl_2$, 1 $MgCl_2$ (pH 7.3 with 95/5% $O_2$/$CO_2$).

For current-clamp recordings, the internal solution contained 124 mM KH2PO₄, 10 mM NaCl, 2 mM MgCl₂, 0.5 mM 1 EGTA, 10 mM glucose, 10 mM HEPES, 2 mM ATP-Na₂, and 0.2 mM GTP-Na (pH 7.3). Current step protocols were used to evoke APs, injecting 500-ms-long depolarizing current steps of increasing amplitude (Δ 5 pA, max 400 pA). Recordings were acquired using a Multiclamp 700 A amplifier (Axon Instruments, Molecular Devices) and a Digidata 1550 (Axon Instruments, Molecular Devices) D/A converter combined with Clampex (Axon Instruments, Molecular Devices). Signals were filtered at 10 kHz, and digitized at 50–100 kHz. Passive properties were calculated using Clampfit (Axon Instruments, Molecular Devices) from the hyperpolarizing steps of the current-clamp step protocol. capacitance was calculated in the current-clamp hyperpolarizing step as follows. First, the resistance was determined as voltage derivative (dV)/DI (voltage/current), and then the cell time constant (tau) was obtained, fitting the voltage changing between baseline and hyperpolarizing plateau. Capacitance was calculated as tau/resistance. Capacitance is the time constant of the voltage between the baseline and the plateau during a hyperpolarizing step. An event was detected as an AP when cross 0 mV and when the rising slope was more than 20 mV/ms. Threshold was defined as the voltage at which the first derivative (dV/dT) reach 10 mV/ms.

For EPSC recording, cells were voltage-clamped at −70 mV. EPSCs were recorded using the same solution used for firing profile characterization. The cell capacitance and series resistance (up to about 75%) were always compensated. Currents were low-pass filtered at 2 kHz and acquired on-line at 5–10 kHz with Molecular Devices hardware and software. Synaptic events were analyzed using MinyAnalysis (Synaptosoft).

## ATAC-seq

Zebrafish: 50 embryos (injected with 200 ng/ul of mRNA) were anaesthetized with Tricaine 160 mg/l at 72 h.p.f. heads were collected in cold PBS and processed for ATAC-Seq analysis. To obtain a homogenous single cell suspension, zebrafish heads were digested using *Adult Brain Dissociation kit, mouse and rat* (Milteny Biotech, 130-107-677). Tissue was first enzymatically digested with papain and Dnase provived by the kit at 37° for 10 min, then mechanically dissociated using a p1000 for 10–20 passages, then digested for another 10 min and 37°. Afterward, a final mechanical dissociation passage was perform using p200 for 10–20 times. Single cells were then filtered through a 70um cell strain to remove debris and centrifuge at 1500 rpm for 5′ at room temperature.

iPSCs, NPCs and neurons: cells were detached from plates with Accutase solution at 37° for 30 min.

For ATAC-seq, 50,000 of cells for each experimental replicate and/or condition were resuspended in lysis buffer (10 mM Tris-HCl, pH 7.4, 10 mM NaCl, 3 mM MgCl₂ and 0.1% IGEPAL CA-630, Sigma aldrich) to perform crude nuclei isolation and centrifuged for 10´at 4° at 0.5 g. Transposition reaction was then performed by resuspending nuclei with Tn5 transposase (Illumina) (2.5 μl for each sample), 2x TD buffer (Illumina) (25 μl for each sample) and nuclease-free water (22.5 μl for each sample). After transposition, DNA was purified using DNA Clean & Concentrator kit (Zymo Research, cat. D4013) and amplified by PCR (NebNext ultra II polymerase, cat. M0544S). After 5 cycles the reaction was stopped, the necessary amount of cycles needed for the final amplification were calculated with qPCR. After amplification, libraries were purified using Ampure XPbeads (Beckman Coulter) selection fragments between 100–1000 bp. Subsequently, samples were sequenced paired-end on NovaSeq6000.

## ChIP-seq

To perform ChIP experiment 200,000 cells for each experimental replicate were used. Cells were cross-linked in 1% formaldehyde for 10 min at room temperature. Cells were lysed in ChIP-lysis Buffer (10 mM Tris-HCl 8.0, 1% SDS, 5 mM EDTA). Subsequently, sonication was performed using a Diagenode Bioruptor Pico sonicator (65 cycles, 30" ON-30" OFF, each cycle). Input was collected before precipitation. Samples (two technical replicates for each ChIP) were then incubated overnight at 4 °C using primary antibodies for H3K27ac and SET in iC buffer from iDeal ChIP-seq kit for Histones (Diagenode, cod. C01010051). The next day, incubation with magnetic beads was performed on a rotator wheel for 3 h at 4°. Beads were subsequently washed using iW1, iW2, iW3, and iW4 buffer (Diagenode). ChIP-DNA was then eluted, decrosslinked for 30 min at 37° then for 4 h at 65 °C and purified using DNA clean & concentrator kit. After quantification, 5 ng of DNA were used to build libraries using NebNext-UltraII kit (NEB, cod.E7645S). Finally, samples were sequenced paired-end Nova-Seq6000. A list of antibodies is provided in the Supplementary Table 1.

## In-situ Hi-C

For in situ Hi-C libraries preparation, 5 millions of either NPCs or neurons, were used per each replicate. The experimental protocol followed Arima HiC kit (A410030) user instructions with minor modifications[53]. Briefly: adherent cells were crosslinked with formaldehyde at 1% concentration at room temperature for 10 min. Digestion was performed with the provided enzyme-mix within intact permeabilized nuclei. For each generated Hi-C library, a minimum of 630 ng up to 2 ug of library were sonicated by Covaris E220 ultrasonicator at the following conditions: 10% Duty factor, 200 cycle, 140 peak, 63 s. Biotin-enriched fragments were size-selected with an average size of 400 bp and subjected to end-repaired with the NEB Next Ultra-II kit (E7645L) and tagged with different Illumina DNA adapters. Libraries were amplified with 6 or 7 cycles following KAPA HyperPrep PCR conditions (07962347001) and verified by q-PCR with KAPA-Q-PCR reagents (07960140001). A first sequencing run of 5 million reads was performed to verify the experimental quality. After passing the quality check, samples were paired-end sequenced on the Nova-Seq6000 platform.

## RNA extraction for RT-qPCR and RNA-seq

RNA from neurons or NPCs was extracted using the TRI Reagent isolation system (Sigma-Aldrich) according to the manufacturer's instructions. For quantitative RT-PCR (qRT-PCR), one microgram of RNA was reverse transcribed using the ImProm-II Reverse Transcription System (Promega). Quantitative RT-PCR was performed in triplicate with custom-designed oligos using the CFX96 Real-Time PCR Detection System (Bio-Rad, USA) using the Titan HotTaq EvaGreen qPCR Mix (BIOATLAS). cDNA was diluted 1:10, was amplified in a 16 μl reaction mixture containing 2 μl of diluted cDNA, 1× Titan HotTaq EvaGreen qPCR Mix (Bioatlas, Estonia), and 0.4 mM of each primer. Analysis of relative expression was performed using the ΔΔCt method, using 18 S rRNA as housekeeping gene and CFX Manager software (Bio-Rad, USA). RNA was sequenced paired by GENEWIZ company (Germany). A list of primers is provided in the Supplementary Table 2.

## Multiome-seq (scATAC+scRNA)

To perform single cell Multiome experiment, fresh cortical tissue was dissected from three different animals for each experimental condition (control: *Rosa26::LoxP-STOP-LoxP-hSETBP1^(G870S/+)*/ Nestin::Cre- and Brain mutant: *Rosa26::LoxP-STOP-LoxP-hSETBP1^(G870S/+)*/Nestin::Cre+) and time point (E14.5 and P2). Cortical tissue was then digested using *Adult Brain Dissociation kit, mouse and rat* (Milteny Biotech, 130-107-677). Tissue was first enzymatically digested with papain and DNAse provided by the kit at 37° for 10 min, then mechanically dissociated using a p1000 for 10–20 passages, then digested for another 10 min and 37°. Afterward, a final mechanical dissociation passage was perform using p200 for 10–20 times. Single cells were then filtered through a 70 nm cell strain to remove debris and centrifuge at 1500 rpm for 5′ at room temperature. After cell counting, cells deriving

from the three different animals for each experimental condition were pulled together and nuclei were extracted following 10x Genomics protocol for Nuclei Isolation from Embryonic Mouse Brain Single Cell Multiome. A total of 1 million cells for each experimental condition were used as a starting point for nuclei extraction. After nuclei counting single cell libraries were prepared using Chromium Next GEM Single Cell Multiome ATAC + Gene Expression Reagent Bundle (10x Genomics, cod. FC51000285). 9000 cells for embryonic samples and 14,000 cells for postnatal samples, were loaded on Chromium controller after Tn5 transposition for single cell library preparation. For each experimental time point a unique RNA-seq and ATAC-seq library was obtained and sequenced on a NovaSeq6000 platform.

**In utero electroporation for neural arborization analysis**

*In utero* electroporation[86] was used to deliver an expression vector into the ventricular zone of mouse embryos at E13.5. The *utero* of E13.5 pregnant female was exposed by midline laparotomy after anesthetization with Avertin (312 mg/kg). DNA plasmid for RFP expression (1 ug) corresponding to 3 ug mixed with 0.03% fast-green dye in PBS was injected in the telencephalic vesicle using a pulled micropipette through the uterine wall and amniotic sac. 7 mm platinum tweezer-style electrodes were placed outside the uterus over the telencephalon and 5 pulses of 40 V, 50 ms in length, were applied at 950 ms intervals by using a BTX square wave electroporator. The uterus was then replaced in the abdomen, the cavity was filled with warm sterile PBS and the abdominal muscle and skin incisions were closed with silk sutures. At post-natal day 10, mice were perfused with 4% PFA and the brain was extracted for subsequent analysis. All procedures were approved by the Italian Ministry of Health and the San Raffaele Scientific Institute Animal Care and Use Committee in accordance with the relevant guidelines and regulations.

**Sholl analysis**

Neuronal cultures derived from NPCs or directly differentiated from iPSCs were transduced with lentiviral vector EF1a-GFP at a low titer at day 4 of differentiation for 1 h in neuronal maturation medium, to obtain sparse GFP cell-labeling. At 4 weeks of differentiation, cells were processed for immunofluorescence analysis as previously described; images of the dendritic tree of double positive GFP + /MAP+ cells were analyzed using Sholl Analysis plugin[87] in ImageJ software (NIH, USA).

**Computational analysis**

**ATAC-seq and ChIP-seq data pre-processing.** All epigenomics dataset were processed using a custom Snakemake (v7.6.1) pipeline, from reads adapter trimming up to peak calling and normalized track generation. FASTQ were first quality checked to evaluate sequence output with using FastQC (Andrews, S. FastQC A Quality Control tool for High Throughput Sequence Data). Reads were trimmed using Trimmomatic[88] (v0.39) and then aligned to reference genome hg38 or danrer11 using Bowtie2[89] using the --very-sensitive option. Non canonical and M chromosomes were removed from Bam files using Samtools[90] (v1.9) and Picard tools ("Picard Toolkit." 2019. Broad Institute, GitHub Repository. https://broadinstitute.github.io/picard) was used to remove PCR optical duplicates, before proceeding with further analysis. Normalized BigWig files for genomics track visualization and for subsequent analysis were generated using deepTools (v3.5.1)[91] 'bamCoverage with the following parameters --normalizeUsing RPKM --binSize 10 --smoothLength 300 – effectiveGenomeSize --ignoreDuplicates --skipNAs –exactScaling'. To obtain a single merged tracks from single experimental replicates for each experimental condition UCSC bigWigMerge was used. Peak calling was performed using MACS2[92] (v2.2.6) from preprocessed Bam files for each experimental replicate from each condition. For ATAC-seq peak calling was performed on the Tn5- corrected single-base insertions using the following parameters '--shift −75 --extsize 150 --nomodel -call-summits

--nolambda --keep-dup all −qvalue 0.01'. For ChIP-seq narrow peaks (H3K27ac), the following parameters were used '-f BAMPE --nomodel --*q*value 0.01 --keep-dup all --call-summits' using Input for each condition as control sample. Finally to obtain a consensus peak set from different replicates for each experimental condition bedtools[93] sort and merge function were used, to avoid the loss of significant signal associated with each condition.

**Chromatin accessibility, Chip-seq heatmaps and profile plots.** To determine regions with major chromatin accessibility/acetylation alterations, the ATAC-seq or H3K27ac peaks found in the control condition of each model, were divided into three different clusters using deeptools (v3.5.1) computeMatrix and plotHeatmap functions by k-means clustering ($n = 3$) based on ATAC or H3K27ac normalized signal of the control and mutant condition. The median of the signal for each region was then plotted in the heatmap in decrescent order. The plorProfile command was instead use to generate density plot of the median signal around peak centers. To annotate the genomic position of each region associated with each cluster we used Chip-Seeker R package[94], setting the promoter region between −10kb /+2 kb from TSS.

**Motif enrichment in open chromatin regions.** To perform motif enrichment inside each open chromatin cluster we used HOMER package scanning +−200 bp from each peak center using findMotifsGenome.pl, adding the following flag '-mask nomotif' to perform just the scanning for known motif.

**Super-enhancers analysis.** To analyze Super-enhancers (SE) we use ROSE (RANK ORDERING OF SUPER- ENHANCERS)[48,49] with stitching parameter set at 12.5 kb. H3K27ac peaks found in NPCs D868D were used as input regions, H3K27ac alignment files were used to calculate signal enrichment for SE.

To determine SE that were affected by alterations in chromatin accessibility and acetylation alterations, we first calculated the number of overlapping ATAC peaks to each SE. Afterward, we calculated the percentage of ATAC peaks overlapping each SE that were associated with the cluster of regions showing a reduce chromatin accessibility in NPCs D868N (cluster 3) on all the peaks overlapping each SE. We then plotted the distribution of SE based on this percentage and selecting those that containing at least 50% of overlapping ATAC peaks associated with cluster 3.

**ATAC-seq integration with annotated chromatin state.** To determine the overlap between our ATAC-seq peaks and known annotated chromatin states we follow a previously published method[44]. Chromatin status annotation from an input 25 states model for different tissues and cell types, lifted over from hg38, were downloaded from Index of /roadmap/data/byFileType/chromhmmSegmentations/ ChmmModels/imputed12marks/jointModel/fi nal (wustl.edu). ATAC peaks of interest were intersect with each tissue or cell line select form the list to obtain the overlap and then plot it. In Supplementary Fig. 1i and Supplementary Fig. 3a, b, f, we compacted some annotations together as following: Promoter (2_PromU, 3_PromD1, 4_PromD2, 22_PromP, 23_PromBiv), Enhancer (13_EnhA1, 14_EnhA2, 15_EnhAF, 16_EnhW1, 17_EnhW2, 18_EnhAc),

Heterochromatin (21_Het), Quiescent (25_Quies), Transcribed (1_TssA, 5_Tx5', 6_Tx, 7_Tx3', 8_TxWk, 9_TxReg, 10_TxEnh5', 11_TxEnh3', 12_TxEnhW), Polycomb (24_ReprPC) and ZNF_Rpts (20_ZNF/Rpts).

A custom R script available in the online code repository was used to calculated fold enrichment and statistical significance of each chromatin state enrichment.

**Chromatin accessibility analysis from iPSCs to NPCs.** To understand the changes happening in chromatin accessibility during the normal

differentiation process, we decided to take advantage of a publicly available ATAC-seq dataset of WT PSCs (GSE122385). We first generated a dataset of the open chromatin regions that were accessible either in PSCs or in our control NPCs cell line (D868D). We used Deseq2[95] to calculate the log2 fold change between the two cell types in each of the regions obtained, then we divided the aforementioned dataset into 4 quartiles, based on the aforementioned log2 fold change. For each different quartile, the ATAC normalized counts associated to PSCs, NPCs D868D and D868N was plotted. To determine where SET affected regions are mainly localized, we intersected the regions present in the cluster 3 with the regions associated with each different quartiles and plot the result (Fig. 2g).

**Hi-C data processing.** To perform raw data QCs and alignment to hg38 reference of Hi-C data, Juicer pipeline (v1.6) was used with default settings[96]. Contact matrices were then generated up to a resolution of 5 kb for each experimental replicate and visualize using. Sample-to-sample correlation was calculated using Hi-C Explorer (v3.7.2)[97] package. First.hic contact matrices files for each samples were converted to H5 format using the function hicConvert, at a resolution of 500 kb. Then sample to sample correlation was calculate using hicCorrelate command. Sample PCA was calculated from.hic contact matrices using fanc (v0.9.23) package setting the maximum distance between bins at 100 kb.

After evaluating quality matrices and sample to sample to correlation we used the Mega script present in the Juicer pipeline to generate one megamap, up to a resolution of 1 kb, for each experimental condition, that were used for subsequent analyses. Arrowhead[53] was used to compute contact domains at a resolution of 5 kb, using default settings. HICCUPS[53] was used to compute loops at a resolution of both 10 and 5 kb which then were merged to create the merged loop list for each sample, using defaults parameters. To plot the relation between interaction frequency and bins distance, we first extract the contact matrix of each sample at 50 kb resolutions using Straw[96]. Data were then plotted using a R custom script.

**Loop strength calculation.** Loop strength was calculated as previously described[98]. First, a merged loop list for the control condition for each experimental model was set as a reference list. Then, the different parameters corresponding to the same exact genomic coordinates in the mutant samples were extracted from the corresponding Hi-C contact map using HICCUPS, submitting the control loop list as a variable of the optional command request loop list. Loop strength for each loop was then calculated as log2(observed/expected bottom left peak) value. To calculate the loop strength changes during neural differentiation a consensus loop list was generated by merging the loop found in NPCs and Neurons computed by HICCUPS for each genotype, then the calculation was carried as mentioned above. Aggregated Peak Analysis (APA) was performed using juicer tools[96] homonymous function and plotted using an R custom script.

**Epigenomics integration with loop anchors.** To integrates loop anchor coordinates with other omics (ATAC-seq, ChIP-seq) a custom R script was used. Every peak falling into any of the two anchors composing each loop were considered as loops associated peak. This set of peaks was then used for subsequent computations and plot generations.

**Chromatin accessibility changes during neural differentiation.** To calculate the regions that show either an increased or a decreased chromatin accessibility during neural differentiation in the control (D868D) or mutant (D868N) condition, we first generated a unique set of open chromatin regions comprising all open regions from NPCs and/or neurons in each genotype. We next used Deseq2[95] to calculate regions that were significantly up or down regulated comparing NPCs

and neurons in the two conditions, setting the following threshold log2FC > 0.5 and p-adjusted value < 0.5.

**Enhancer-promoter contact calculation.** To associate a gene to each ATAC-seq peaks in each cell type, we analyzed the corresponding Hi-C matrix contact frequency data using a custom R-script. Briefly, the Hi-C matrix was binned at 50 kb. The bins containing at least a gene promoter were identified using the hg38 human TSS dataset from the ChipPeakAnno[99] R package. Afterward, for each gene, all the bins interacting with the bin containing their cognate promoter were isolated as regulatory regions containing bins associated to that specific gene. ATAC-peaks falling into those bins were then identified and they were deemed as regulatory regions of that specific gene. If an ATAC peak was falling into a bin interacting with more than one promoter, it was assigned to the promoter with the highest interaction frequency obtained from the Hi-C matrix.

**RNA-seq analysis.** NPCs RNA-seq data and analysis were obtained from our previously published work[39] (GSE171266). We further added one replicate for NPCs control and mutant. RNA sequencing from neurons were processed as previously published[39]. FASTQ reads were quality checked and adapter trimmed with FastQC (Andrews, S. FastQC A Quality Control tool for High Throughput Sequence Data). High quality trimmed reads were mapped to the hg38 reference genome with the STAR aligner[100] using the latest GENCODE main annotation file[101]. Differential gene expression was calculated with DESeq2, using a p-adjusted value cut-off of 0.1[95]. Functional enrichment was performed using gprofiler2 R package for all Gene expression data.

**scMultiome data pre-processing.** Raw sequencing data deriving from both the scRNA and scATAC libraries were aligned to the GRCm38 (mm10) reference genome and quantified using 'cellranger-arc count' (10x Genomics, v.2.0.0). Sample integration was performed using 'cellranger-arc aggr'. For scRNA data further data processing was performed using R package Seurat[102] and for scATAC, R package ArchR[68] was used. We proceed to isolate a unique high-quality dataset of cells (n = 29,372) using specific cutoff for both RNA and ATAC quality.

**scMultiome clustering.** Cells clustering was based on the scRNA dataset using the following strategy. Only the top 2000 highly variable genes, as identified by the function "FindVariableGenes" were considered. Variable genes were then used to perform principal component (PC) analysis. We selected the PCs to be used for downstream analyses by evaluating the "PCEl-bowPlot" and the "JackStrawPlot". We used the first 15 PCs. We identified clusters using the function "FindClusters", which exploits a SNN modularity optimization clustering algorithm (at Resolution = 0.7). Clusters visualization was performed using the uniform manifold approximation and projection (UMAP) dimensionality reduction[103]. We used the manual inspection of marker genes determined using the "FindAllMarkers" function for cluster identification. This function determines which genes, that are expressed in at least three cells, are enriched in every clustering using log2FC threshold values of 0.25 and 0.05 of adjusted P value (FDR). All steps were performed using Seurat[103]. Clusters with a similar marker profile were fused together to simplify subsequent analysis.

Visualization of clusters based on scATAC dataset was performed with the following procedure. First Latent Semantic Indexing (LSI) algorithm was applied to scATAC data, as previously published[104], to reduce the dimensionality of the dataset. Then UMAP algorithm was used to physically visualized the data, maintaining the same clusters obtained from the scRNA dataset. All the above steps were performed using ArchR[68].

Correlation between Predicted gene expression based on chromatin accessibility and the gene expression data was performed using

correlateMatrices function from ArchR and plotted using Enhanced-Volcano R package.

**Peak calling strategy for scATAC dataset.** Due to the presence in our dataset of two different genotypes, we consider that the approach for peak calling normally used by ArchR, would cause a great loss of information, especially genotype specific differences. To solve this issue, we decided to maintain the normal workflow of peak calling used by ArchR, combining macs2[92] and the iterative overlap[105] approach, but instead of using the whole cells dataset, we performed the peak calling on the cells associated to each clusters within genotype. Thus, this approach allowed us to obtain a highly genotype specific peak set for each cluster. These peak lists were then used accordingly for any subsequent analysis.

**RNA-velocity and pseudotime.** For all the analyses that follow, we used the UMAP and the clusters obtained from the scRNA data. Velocyto[67] was used to obtain spliced vs unspliced specific count data for each sample. Integration of each separate loom file was then performed using scVelo[106]. Average velocity value associated to each cluster were then plotted as heatmap distinguishing between control and mutant.

Pseudotime calculation based on scATAC data was performed using ArchR[68]. Cell trajectories of specific cellular lineages were preselected, then the pseudotime value for each cell was plotted on the selected UMAP. The analysis was performed separately for each genotype.

This analysis was also used to visualize the expression of specific genes along the pseudotime (Fig. 6h, Supplementary Fig. 9e) and to understand which ATAC-peaks were mostly associated with specific developmental trajectories (Fig. 6g). For the latter analysis, ArchR was used to calculate which peaks were strongly associated, thus differentially accessible, along the trajectory of interest. For each heatmap, the accessibility Z-score (peak accessibility) across pseudotime for each peak was plotted. Only highly correlated peaks, with a variance quantile cut-off above 0.9, were plotted.

Pseudotime calculation based on scRNA data was also performed, using Monocole3[107] (Fig. 6k). For this specific analysis ExN_DL and ExN_UL clusters were merged to obtain a single neuronal cluster.

**Enhancer-promoter contact analysis in scMultiome.** To determine which gene was associated to each peak we used the distance criteria, using ChIPSeeker[95]. We elected to not use classical co-accessibility method such as Cicero[108], due to our necessity to keep the analysis separate between the two genotypes. By annotating the peak by distance, we avoided the situation in which the same peak might be associated to two different genes. To analyze the differential enhancers usage between the two condition we design a custom R script which can discriminate for each peak associated to a gene if the peak was shared or not between the two genotypes.

**scATAC track generation.** Merged BigWig files for each cluster were generated using ArhR function 'getGroupBW', normalizing the signal using the number of reads inside TSS.

**Statistics and reproducibility.** Data in figure panels reflect several independent experiments performed on different days. Each experiment has been performed at least 3 times separately, except for ChIP-seq where 2 indipendent experiment were performed for each condition. For animal experiments each data point in a graph represent an independent animal. No data were excluded. No statistical methods were used to predetermine sample size in other experiments. Samples were not subject to randomization but were assigned to experimental groups based on their genotype. Data are expressed as mean ± standard error (SEM) and significance was set at $P < 0.05$ (see each figure for details). Statistical details are indicated in the figure legends. Differences between means were analyzed using the Student's two-sided unpaired $t$ test, two sided Wilcoxon-test and one-way or two-way ANOVA depending on the number of groups and variables in each experiment. Data were then submitted to multiple comparison using Tukey post hoc test using. Statistics was computed using GraphPad Prism software. Genomic statistical data analysis was all performed in the R software environment.

**Reporting summary**
Further information on research design is available in the Nature Portfolio Reporting Summary linked to this article.

## Data availability
The ATAC-seq, ChIP-seq, Hi-C, RNA-seq and scMultiome data generated in this study have been deposited in the Gene Expression Omnibus (GEO) database in the are available at GSE212252. NPCs RNA-seq data are available at GSE171266. Control PSC ATAC-seq were obtained from GSE108248 https://docs.github.com/en/repositories/archiving-a-github-repository/referencing-and-citing-content. NPCs and NPC-derived neurons Hi-C control dataset[54] used for this publication were obtained from NIMH Repository & Genomics Resource, a centralized national biorepository for genetic studies of psychiatric disorders. The raw number associated with bar plots pertaining the associated figures, ATAC-seq peaks, ChIP-seq peaks, differentially expressed genes, RNA normalized counts, single cells matrices, functional enrichment results generated in the study are available in the source data and supplementary information files of this work.

## Code availability
All original codes used for Data processing in this work are available on Github or Zenodo[109].

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

## Acknowledgements

We thank Prof. Alexander Hoischen for sharing the fibroblasts from SGS patients, them, and their families. We thank SGS foundation for continuous support. We are indebted with Dr. Bruno Di Stefano, Dr. Dario Bonanomi, and the members of Broccoli's lab for the helpful suggestions. We thank the San Raffaele The Advanced Light and Electron Microscopy BioImaging Center (ALEMBIC) for help in image acquisition. This work was supported by the Italian Ministry of Health (GR-2019-12370949) (A.S.), Fondazione Regionale per la Ricerca Biomedica (Regione Lombardia), European Joint Program on Rare Disease Project TREAT-SGS (EJPRD20-008), GA 825575 (A.S.) and. by a Career Development Award from the Giovanni Armenise-Harvard Foundation (J.-M.C.). Data used as benchmark control in this work were obtained from NIMH Repository & Genomics Resource, a centralized national biorepository for genetic studies of psychiatric disorders, and were generated as part of the PsychENCODE Consortium, supported by: U01DA048279, U01MH103339, U01MH103340, U01MH103346, U01MH103365, U01MH103392, U01MH116438, U01MH116441, U01MH116442, U01MH116488, U01MH116489, U01MH116492, U01MH122590, U01MH122591, U01MH122592, U01MH122849, U01MH122678, U01MH122681, U01MH116487, U01MH122509, R01MH094714, R01MH105472, R01MH105898, R01MH109677, R01MH109715, R01MH110905, R01MH110920, R01MH110921, R01MH110926, R01MH110927, R01MH110928, R01MH111721, R01MH117291, R01MH117292, R01MH117293, R21MH102791, R21MH103877, R21MH105853, R21MH105881, R21MH109956, R56MH114899, R56MH114901, R56MH114911, R01MH125516, and P50MH106934 awarded to: Alexej Abyzov, Nadav Ahituv, Schahram Akbarian, Alexander Arguello, Lora Bingaman, Kristin Brennand, Andrew Chess, Gregory Cooper, Gregory Crawford, Stella Dracheva, Peggy Farnham, Mark Gerstein, Daniel Geschwind, Fernando Goes, Vahram Haroutunian, Thomas M. Hyde, Andrew Jaffe, Peng Jin, Manolis Kellis, Joel Kleinman, James A. Knowles, Arnold Kriegstein, Chunyu Liu, Keri Martinowich, Eran Mukamel, Richard Myers, Charles Nemeroff, Mette Peters, Dalila Pinto, Katherine Pollard, Kerry Ressler, Panos Roussos, Stephan Sanders, Nenad Sestan, Pamela Sklar, Nick Sokol, Matthew State, Jason Stein, Patrick Sullivan, Flora Vaccarino, Stephen Warren, Daniel Weinberger, Sherman Weissman, Zhiping Weng, Kevin White, A. Jeremy Willsey, Hyejung Won, and Peter Zandi.

## Author contributions

A.S. conceived the study. M.Z., and A.S. designed the experiments. M.Z., F.B., M.V., S.B., C.B., M.B., L.B. L.P., D.D'A., S.S., and E.A. performed the experiments. M.Z., L.Massimino, E.B., I.M., and L.A.L. performed computational analyses. L.Mologni, G.C., F.U., J.-M.C., E.A., R.P., E.M., and V.B. provided materials and advice. A.S. wrote the manuscript with input from all authors.

## Competing interests

The authors declare no competing interests.
