## [Peer Review File · Nature Communications]

Balanced SET levels favor the correct enhancer repertoire during cell fate acquisitionREVIEWER COMMENTS

Reviewer #1 (Remarks to the Author):

With great pleasure I have read the manuscript by Zaghi et al, that explores the role of different SET levels in gene regulation, using an impressive combination of multi-omics assessment (RNA-seq, ATAC-seq, ChIP-seq, chromatin conformation analysis) in a plethora of model systems (in vivo in zebrafish and a mouse model (incl single cell analysis), and in both transgenic as well as patient derived iPSC cells incl. isogenic controls, neural differentiations and pharmacological interventions), with all experiments being well presented and incl. all relevant controls. In my opinion, the authors should have tried even a journal with a higher impact factor from the Nature family. I only have a few minor comments, and in my opinion this work deserves rapid publication. General note: the term enhancer warrants some caution; I would suggest that the authors call those regions which are not functionally validated and simply based on epigenome analysis (e.g., ChIP-seq for H3K27ac or ATAC-seq) putative enhancers, as it is well-known that not all these regions are really active regulatory elements. Therefore simplifying them as enhancers might confuse the literature

I might overlook it, but I do not see a description in the methods of the source of the patient derived iPSCs, other than an acknowledgement to the Nijmegen group. Their origin, way of reprogramming as well as consent including the IRB approval for patient iPSC reprogramming under which these cells were derived should be described, especially given that if these cells come from the Netherlands, IRBs here usually require a specific written consent from patients/legal guardians for patient derived cells to be used in iPSC based studies.

Minor comments:

Line 11, page 4: "over-binding" not sure I would use this nomenclature, consider revising?

Line 34, page 8: ATACseq instead of ATAC-seq

There is a mistake for the tile of Figure1 i. It should be "figure1 h", not "figue1i"

Reviewer #2 (Remarks to the Author):

Study carried by Zaghi investigated the role of SET in chromatin accessibility, histone acetylation and 3D chromatin conformation in the context of neuronal development and SGS, a disease caused by a new spontaneous mutation in the SETBP1. By quantitative alterations of SETBP and SET protein levels in human, mouse and drosophila cells, they characterized the molecular functions of SET and SETBP1 in establishment of active chromatin states, histone acetylation and configuration of proper enhancer accessibility, and enhancer-promoter interactions of genes important in cell-fate and differentiation. Using SGS models, the team generated genome-wide chromatin states, histone modification, 3D chromatin conformation, RNA expression and single-cell multiome data from iPSCs derived NPCs as well as NPC-derived neurons from two independent SGS patients (D868N and I871T) to decode chromatin responses to SET accumulation by comparing with their corresponding isogenic controls (D868D and I871I). Lastly, these changes in open chromatin, H3K27ac modification and enhancer usage in promoter activation were associated with gene expression and related neuron-associated developmental abnormality.

I found the study was well presented, the results were supported by well-designed experiments and solid data and the significance was well articulated. The datasets generated are valuable for the genes and pathways regulated by SET/SETBP1 in neuronal development and SGS disease. The following comments can be addressed to further improve the quality of this manuscript:

- The high levels of SET proteins in SGS patient-derived mutant NPCs were not shown (extended data fig. 1e). Western blot from reciprocal co-IP only showed slight increases in SET (Figure 1a).
- The chromatin accessibility changes in D868N and I871T NPCs could be better demonstrated as correlation plots of ATAC-seq signals between D868D and D868N cells similar to the plots shown in Extended Data Fig. 1f
- The enrichment of ATAC-seq peaks found in cluster 3 of D868D (figure 2a) for distal enhancers

was not shown. Different classes of ChromHMM states were found to have elevated levels but the enrichment folds of different classes should be determined.

- How can the authors explain the large differences of the numbers of ATAC-seq peaks between NPCs D868D (89437) and I871I (66721)? Both are isogenic wild type controls (Extended data Fig. 1i). I871I has even fewer ATAC-seq peaks than these found in D868N (73585).
- What is the genomic nature that differentiates cluster 1 from cluster 2?
- Among the 89437 ATAC-seq peaks found in D868D NPC cells, ~ 2/3 of them are in cluster 2, yet, only 44% of SEs (n=611) contain $\geq 50\%$ of the cluster 3 ATAC-seq peaks. Looking at figure 2e, there is a set of SEs with little or no overlap with cluster 3 peaks. This indicated that cluster 3 and SEs could be quite distinct. If this is the case, what is the relevance of SEs in SET function? Can author perform the same analysis like figure 2e using clusters 1 and 2 peaks?
- How do the 60203 ATAC-seq peaks detected in cluster 3 of D868D (Figure 2a) distribute among different quartile of differential ATAC-seq peaks found between iPSC and NPC WT cells? Is the difference between different quartiles significant? If so, I think the effect of SET accumulation on distal regulatory elements during neural development can be clearly established.
- The authors claimed strong concordance between transcriptome (scRNA-seq) and chromatin accessibility (scATAC-seq) based on the UMAP clusters (Fig. 6e and Extended Data Fig. 6g). What is the level of concordance and how was it determined?

Minor points (most of them are typo or editing issues):

- Extended Data Fig. 1l legend n should be 153,930, not 136,813.
- Fig. 1i that was referred in main text (line 10 in the 2nd page of the result section) and Fig file should actually be Fig. 1h.
- Extended Data Fig. 2d legend "cluster1=11,616, cluster2=29,929, cluster3=84,192 found in zebrafish". These numbers are from iPSCs in Extended Data Fig. 2c.
- Figure 2d legend SE number n=1,367, should be n=1,387.
- Figure 4b legend "mutant light blue D868N n=12,486" should be n=13,176.
- Extended Data Fig. 6e legend. The description is incorrect. It was Nestin:Cre, not CMV:Cre, used here.

Reviewer #3 (Remarks to the Author):

The manuscript by Zaghi and colleagues investigated the molecular consequences of the abnormal accumulation of the SET protein, detected in a number of diseases, on genomic regulation during neuronal specification and neurodevelopment. They used a wide range of techniques to study the alterations in chromatin accessibility, histone acetylation, use of genomic elements in both in vitro and in vivo models. The authors reported that abnormally high level of SET protein was associated with loss of histone acetylation (H3K27ac) and chromatin accessibility in NPCs, NPC-derived neurons and directly reprogrammed neurons, which is correlated with weakened chromatin looping, resulting in aberrant expression of the associated genes. The author proposed a balanced level of SET is important for correct cell fate commitment where imbalanced SET level led to maladaptive mechanisms that reduced the utilization of distal elements or enhancers. Instead, an alternative set of distal regulatory elements that is closer to the promoters were used for transcription of genes that are important during neuronal specification. They showed that inhibition of histone deacetylation could alleviate the neuronal morphological impairment although it is unclear whether this rescued or restored the electrophysiological properties. Finally, the author reported differential chromatin usage in an in vivo model of SGS showing delayed neuronal development. The author proposed abnormal accumulation of SET protein as a major pathophysiological mechanism in diseases where elevated SET level is detected including SGS, neurodevelopmental disorders, myeloproliferative diseases and cancer.

The work is interesting and comprehensive, with a massive amount of data. Sufficient details were provided in the methods for the work to be reproduced. Overall, the manuscript is well-written and provides novel insights into the pathophysiological mechanisms of neurodevelopmental disorders where accumulation of SET was detected.

However, there are some issues that should be addressed, as detailed below.

Major comments:

1. While the study provided evidence for accumulation of SET protein affect neuronal specification and delayed neurodevelopment in in vitro and in vivo models, the authors made a strong statement that differential cis-regulation caused by elevated SET level as a key contributing factor for SET-related disorders including SGS, neurodevelopmental disorders, myeloproliferative diseases and cancer. However, as stated by the authors the Introduction and Discussion, "direct evidence of chromatic defects and their eventual connection has never been investigated." While it is an intriguing hypothesis, the authors might want to tune down the statement as the manuscript did not report evidence of differential cis-regulation in other SET-related disorders, e.g. cancer. Otherwise, citations of the relevant publications should be included and discussed.
2. While it is interesting to see that SET accumulation was shown in SGS NPCs, iPSCs and NPCs overexpressing SET and in zebrafish infected with SET mRNA, and that impairments in H3K27ac, chromatin accessibility and chromatin loop contacts have been shown in a combination of models, as the authors also pointed out in the Discussion that the contribution of SETBP1 cannot be overlooked, since it is a chromatin remodeller and transcription factor itself, which was also shown in the authors' previous publications. SETBP1 carrying an SGS variant is likely to affect chromatin functions and can cause these impairments, and the extent of contribution of abnormal SETBP1 and SET level to these impairments is unknown. Without showing that reducing SETBP1 and SET protein level could reverse abnormal cis-regulation reported, the author could not be sure that SETBP1 did not contribute to these impairments. The authors should therefore tune down their conclusion about SET and elaborate more on the plausible contribution of SGS SETBP1 in the Discussion. Otherwise, the authors should add an experiment addressing these points.

Minor:

(The line number restarts on each page but there is no page number in the manuscript pdf so I will refer to the section and paragraph in the rest of the comments.)

Results section:

- In the second paragraph of "SET levels influence histone acetylation in multiple models":
 - o Line 7: it was not clear on which samples ChIP-seq with SET antibody was performed. Is there Was there differential binding targets of SET in different conditions or only the strength was different?
 - o In the following sentence in line 7-10, "At the genome-wide level, all SEThigh conditions examined were featured by consistently less called peaks compared to the relative controls (Extended Data Fig. 1i), while all pairwise comparisons were similar in term of accessibility of the identified peaks (Fig. 1g, i, Extended data Fig. 1j-k).", it was not clear that it was referring to the ATAC-seq peak, please specify in the text.
- In the second paragraph of "SGS NPCs display rearranged chromatin topography", lines 23-26: "A similar pattern was observed in the RIMS4 locus (synaptic protein connected with autism cases). Notably, both genes resulted reduced in mutant cells in relation to the controls (Fig. 3i)." It should be (Extended Data Fig. 3g) instead of (Fig. 3i).
- In the first paragraph of "Neuronal development is regulated in an alternative way in SGS",
 - o Line 8, "A straight comparison of open chromatin peaks between control NPCs... during neuronal differentiation in vitro (Fig. 4a)." This sentence is incomplete, rephrase.
 - o Line 12, "Anyway, at the qualitative level, the regions between the two genotypes were different, with only 4,488 and 2,796 peaks that gained and lost accessibility in both conditions, respectively (Fig. 4b, Extended Data Fig. 4c, Supplementary Table 9)."  it is unclear which "both conditions" were referred to, please specify.
- In the second paragraph of "HDAC inhibitor fosters the maturation of SGS neurons", line 9, "which successfully restored the H3 acetylation levels in SGS neurons (Extended Data Fig. 5a)."  but the figure showed data of SGS NPCs, please correct.
- In the third paragraph of "HDAC inhibitor fosters the maturation of SGS neurons", "These results sustain the model by which chromatin defects, secondary to high levels of SET, are responsible for defective neuronal differentiation in SGS."  The authors cannot conclude from these data that high SET level is the only factor responsible for defective neuronal differentiation in SGS without an experiment reducing SET level and see if the phenotype was rescued, especially when the impairments were only alleviated.
- The authors first showed that there was earlier opening of neuron-specific TFBS and persistent opening of TFBE typical to NPCs and persistent closure of neuronal-specific ones in SGS NPCs and

neurons but delayed neuronal maturation was shown in SGS whole mouse brain. In the Discussion, it was also mentioned that there was early sign of degeneration with DNA damage in SGS organoids as shown in Banfi et al., 2021, are the mechanisms shown, i.e. delayed neuronal maturation and early degeneration of neurons mutually exclusive, or it is a combination of both? Please elaborate in the Discussion.

- Legends of Fig 4h, the figure showed data of SEMA3A but legends said RSPH3.
- Legends of Fig 6l, should be E14.5 instead of E24.5
- Legends of Extended Data Fig. 6e, "CMV::Cre and Rosa26-LoxP-STOP-LoxP-hSETBP1G870S" should be "Nestin::Cre" if it is for getting whole brain specific mutant?
- Legends of Extended Data Fig. 6g, E14.5 instead of E145.

Point by Point Response about our MS NCOMMS-22-43699-T by Zaghi et al.

Reviewer #1 (Remarks to the Author):

With great pleasure I have read the manuscript by Zaghi et al, that explores the role of different SET levels in gene regulation, using an impressive combination of multi-omics assessment (RNA-seq, ATAC-seq, ChIP-seq, chromatin conformation analysis) in a plethora of model systems (in vivo in zebrafish and a mouse model (incl single cell analysis), and in both transgenic as well as patient derived iPSC cells incl. isogenic controls, neural differentiations and pharmacological interventions), with all experiments being well presented and incl. all relevant controls. In my opinion, the authors should have tried even a journal with a higher impact factor from the Nature family. I only have a few minor comments, and in my opinion this work deserves rapid publication.

We really thank the reviewer for her/his generous comments on our work and for the suggestions that have improved the quality of the revised manuscript.

Here below you can find a point-by-point response to the issues addressed and please note that we highlighted the changes in the manuscript using red sentences.

We hope that the new version of the manuscript has eliminated the weaknesses highlighted by this reviewer.

General note: the term enhancer warrants some caution; I would suggest that the authors call those regions which are not functionally validated and simply based on epigenome analysis (e.g., ChIP-seq for H3K27ac or ATAC-seq) putative enhancers, as it is well-known that not all these regions are really active regulatory elements. Therefore, simplifying them as enhancers might confuse the literature

We thank the reviewer for pointing this out. We have now indicated them as putative enhancers.

I might overlook it, but I do not see a description in the methods of the source of the patient derived iPSCs, other than an acknowledgement to the Nijmegen group. Their origin, way of reprogramming as well as consent including the IRB approval for patient iPSC reprogramming under which these cells were derived should be described, especially given that if these cells come from the Netherlands, IRBs here usually require a specific written consent from patients/legal guardians for patient derived cells to be used in iPSC based studies.

The iPSC lines used in this work have already been described in our previous work (Banfi et al., Nat Commun, 2021, doi: 10.1038/s41467-021-24391-3). We have also deposited the information associated with these lines in the repository hPSC^{reg} (<https://hpscereg.eu>) which released us certificates (#OSRi009-A, OSRi009-A-1, OSRi010-A, OSRi010-A-1). This information is now included in the manuscript.

Minor comments:

Line 11, page 4: "over-binding" not sure I would use this nomenclature, consider revising?

Line 34, page 8: ATACseq instead of ATAC-seq

There is a mistake for the tile of Figure1 i. It should be "figure1 h", not "figure1i"

We have corrected the indicated points in the new version of the manuscript

Reviewer #2 (Remarks to the Author):

Study carried by Zaghi investigated the role of SET in chromatin accessibility, histone acetylation and 3D chromatin conformation in the context of neuronal development and SGS, a disease caused by a new spontaneous mutation in the SETBP1. By quantitative alterations of SETBP and SET protein levels in human, mouse and drosophila cells, they characterized the molecular functions of SET and SETBP1 in establishment of active chromatin states, histone acetylation and configuration of proper enhancer accessibility, and enhancer-promoter interactions of genes important in cell-fate and differentiation. Using SGS models, the team generated genome-wide chromatin states, histone modification, 3D chromatin conformation, RNA expression and single-cell multiome data from iPSCs derived NPCs as well as NPC-derived neurons from two independent SGS patients (D868N and I871T) to decode chromatin responses to SET accumulation by comparing with their corresponding isogenic controls (D868D and I871I). Lastly, these changes in open chromatin, H3K27ac modification and enhancer usage in promoter activation were associated with gene expression and related neuron-associated developmental abnormality.

I found the study was well presented, the results were supported by well-designed experiments and solid data and the significance was well articulated. The datasets generated are valuable for the genes and pathways regulated by SET/SETBP1 in neuronal development and SGS disease.

We really thank the reviewer for her/his generous comments on our work and for the suggestions that have improved the quality of the revised manuscript. Indeed, following the reviewer's recommendations, we added this set of new results:

- correlation between chromatin accessibility levels in control and mutant pairs;
- Statistical significance of ChromHMM analysis;
- better evaluation of genomic differences among the clusters of ATAC results;
- better evaluation of superenhancers;
- statistics about cluster distribution among quartiles;
- correlation between RNA and ATAC in multiome experiment;

Here below you can find a point-by-point response to the issues addressed and please note that we highlighted the changes in the manuscript using red sentences.

We hope that the new version of the manuscript has eliminated the weaknesses highlighted by this reviewer.

The following comments can be addressed to further improve the quality of this manuscript:

- The high levels of SET proteins in SGS patient-derived mutant NPCs were not shown (extended data fig. 1e). Western blot from reciprocal co-IP only showed slight increases in SET (Figure 1a).

We have shown in our previous work, Banfi et al., Nat Commun 2021 (doi: 10.1038/s41467-021-24391-3), that these NPC lines show a significant SET accumulation. However, for clarity, we added a western blot showing the increase of SET protein (Supplementary fig.1f in the new version of the manuscript).

- The chromatin accessibility changes in D868N and I871T NPCs could be better demonstrated as correlation plots of ATAC-seq signals between D868D and D868N cells similar to the plots shown in Extended Data Fig. 1f

We appreciate the suggestion of the reviewer. We added the requested correlation plot of ATAC-seq signal between D868N and I871T lines (Supplementary fig.1g in the new version of the manuscript). We also performed the same analysis between the two control lines (D868D and I871I) (Supplementary fig.1g in the new version of the manuscript). The degree of correlation between these two pairs is highly similar, suggesting for a comparable effect of SGS mutation in both genotypes.

- The enrichment of ATAC-seq peaks found in cluster 3 of D868D (figure 2a) for distal enhancers was not shown. Different classes of ChromHMM states were found to have elevated levels but the enrichment folds of different classes should be determined.

We thank you the reviewer for the important suggestion. In the new version of the manuscript, we have determined the fold enrichment and statistical significance for each chromHMM classes in each different cluster (new Supplementary fig. 3b). With these new data, we confirmed that the enrichment for the enhancer classes is stronger in cluster3 while the enrichment for the promoters and TSS features is higher in cluster1 and cluster2.

- How can the authors explain the large differences of the numbers of ATAC-seq peaks between NPCs D868D (89437) and I871I (66721)? Both are isogenic wild type controls (Extended data Fig. 1i). I871I has even fewer ATAC-seq peaks than these found in D868N (73585).

We acknowledge the observation by the reviewer as fair point. However, individual differences reflecting different donor to donor variability may explain the observed differences. These observations have been made also in ATAC-seq experiments in other recently published works using hNPCs or brain regions (de la Torre-Ubieta *et al*, 2018, DOI: 10.1016/j.cell.2017.12.014; Corces *et al*, 2020, DOI: 10.1038/s41588-020-00721-x). However, as suggested by the aforementioned line-to-line correlations (Supplementary fig.1g in the new version of the manuscript), the presence of SGS mutations induces a comparable effect regardless the genetic background.

- What is the genomic nature that differentiates cluster 1 from cluster 2?

This is an interesting point. We performed additional analysis to try to answer this question and better characterize the ATAC-seq clusters that we added in Supplementary Fig.3.

We started by associating each cluster peak to a gene by proximity. The average expression of the genes associated with each cluster retrieving that those in cluster 1 are generally more expressed than the others (Supplementary Fig. 3c), indicating that cluster 1 is associated with genes that are presumably strongly active in NPC state that those poised for activation in following cell stages. Cluster 1 peaks resulted closer to TSS than cluster 2 and cluster 3 peaks (Supplementary Fig. 3d), indicating that even though the first two contains a larger fraction of promoters than cluster 3. Cluster 2 still contains a proportion of putative regulatory regions, as already highlighted by

the chromHMM chromatin state enrichment retrieved from the publicly available dataset (Supplementary Fig. 3a, b), which indicated that in comparison to cluster 1 clusters 2 and 3 contains a bigger share of regions marked by H3K4me1.

Once confirmed that the main difference among clusters is the proportion of the distal regions, we tried to find out whether the distal peaks in the cluster 1 and 2 (not affected in SGS) and those belonging to cluster 3 (affected) have differential genomic features. However, we did not find gross differences when compared with chromatin states from available neural cells and tissues (Supplementary Fig. 3f). This suggests that the SGS condition affects several but not all distal regulatory elements following a yet unknown mechanism that needs to be further explored in future.

- Among the 89437 ATAC-seq peaks found in D868D NPC cells, ~ 2/3 of them are in cluster 3, yet, only 44% of SEs (n=611) contain $\geq 50\%$ of the cluster 3 ATAC-seq peaks. Looking at figure 2e, there is a set of SEs with little or no overlap with cluster 3 peaks. This indicated that cluster 3 and SEs could be quite distinct. If this is the case, what is the relevance of SEs in SET function? Can author perform the same analysis like figure 2e using clusters 1 and 2 peaks?

The analysis of SE was originally proposed as an attempt to better describe our results. The expectation was that a subgroup of these SE may be affected in SGS as indicated in our original MS. For sure these affected SE do not represent neither the only or the majority of the targets of the SET function.

As suggested, we expanded our analysis by intersecting cluster 1 & 2 with our retrieved SEs dataset. As expected, a minor portion of SE is associated ($\geq 50\%$ peaks in cluster) to each cluster. These SE do not display defects in H3K27ac and even a higher chromatin accessibility in mutant than in control NPCs.

- How do the 60203 ATAC-seq peaks detected in cluster 3 of D868D (Figure 2a) distribute among different quartile of differential ATAC-seq peaks found between iPSC and NPC WT cells? Is the difference between different quartiles significant? If so, I think the effect of SET accumulation on distal regulatory elements during neural development can be clearly established.

We agree that a statistical test would help to draw stronger conclusions. Indeed, after the execution of the proportion test (Bain *et al.*, 1987, DOI: 10.2307/2532587), we do see that the distribution between different quartiles of cluster 3 peaks is significant. This information is now added in the Fig. 2g.

- The authors claimed strong concordance between transcriptome (scRNA-seq) and chromatin accessibility (scATAC-seq) based on the UMAP clusters (Fig. 6e and Extended Data Fig. 6g). What is the level of concordance and how was it determined?

We apologize the reviewer for our initial simplistic observation. Having seen in the ATAC-seq data space, a recapitulation of the RNA-based cell clustering, we suggested a strong concordance between the two datasets. However, as the reviewer indicated, this is not necessarily the case. To properly assess this issue, we calculated for each expressed gene in our multiome experiment, the correlation between the real expression (RNA-seq data) and the predicted expression value based on chromatin accessibility level of the associated peaks (Granja *et al.*, 2021, 10.1038/s41588-021-

00790-6). This analysis showed a positive correlation for a large part of the genes (Supplementary Fig. 9c), indicating a good concordance between the two modalities. Importantly, the key markers, that we used as predictor of cell cluster identity, are among the group of genes with high and significant correlation level.

We recognize, for a group of genes, no correlation, or an anti-correlation between the two modalities. This is not a complete surprise, because although chromatin accessibility and transcription are usually correlated, there are instances in which open regions do not show any sign of transcription, at least transiently. As an example, it has been demonstrated that some regulatory region opening may precede mRNA expression (Ma *et al*, 2020, DOI: 10.1016/j.cell.2020.09.056).

Minor points (most of them are typo or editing issues):

- Extended Data Fig. 1l legend n should be 153,930, not 136,813.
- Fig. 1i that was referred in main text (line 10 in the 2nd page of the result section) and Fig file should actually be Fig. 1h.
- Extended Data Fig. 2d legend “cluster1=11,616, cluster2=29,929, cluster3=84,192 found in zebrafish”. These numbers are from iPSCs in Extended Data Fig. 2c.
- Figure 2d legend SE number n=1,367, should be n=1,387.
- Figure 4b legend “mutant light blue D868N n=12,486” should be n=13,176.
- Extended Data Fig. 6e legend. The description is incorrect. It was Nestin:Cre, not CMV:Cre, used here.

We thank the reviewer for these suggestions, we amended the text accordingly.

Reviewer #3 (Remarks to the Author):

The manuscript by Zaghi and colleagues investigated the molecular consequences of the abnormal accumulation of the SET protein, detected in a number of diseases, on genomic regulation during neuronal specification and neurodevelopment. They used a wide range of techniques to study the alterations in chromatin accessibility, histone acetylation, use of genomic elements in both in vitro and in vivo models. The authors reported that abnormally high level of SET protein was associated with loss of histone acetylation (H3K27ac) and chromatin accessibility in NPCs, NPC-derived neurons and directly reprogrammed neurons, which is correlated with weakened chromatin looping, resulting in aberrant expression of the associated genes. The author proposed a balanced level of SET is important for correct cell fate commitment where imbalanced SET level led to maladaptive mechanisms that reduced the utilization of distal elements or enhancers. Instead, an alternative set of distal regulatory elements that is closer to the promoters were used for transcription of genes that are important during neuronal specification. They showed that inhibition of histone deacetylation could alleviate the neuronal morphological impairment although it is unclear whether this rescued or restored the electrophysiological properties. Finally, the author reported differential chromatin usage in an in vivo model of SGS showing delayed neuronal development. The author proposed abnormal accumulation of SET protein as a major pathophysiological mechanism in diseases where elevated SET level is detected including SGS, neurodevelopmental disorders, myeloproliferative diseases, and cancer.

The work is interesting and comprehensive, with a massive amount of data. Sufficient details were provided in the methods for the work to be reproduced. Overall, the

manuscript is well-written and provides novel insights into the pathophysiological mechanisms of neurodevelopmental disorders where accumulation of SET was detected.

We really thank the reviewer for her/his generous comments on our work and for the suggestions that have improved the quality of the revised manuscript. Indeed, following the reviewer's recommendations, we amended the text in accordance with the demand of tune down some claims (see details below).

Here below you can find a point-by-point response to the issues addressed and please note that we highlighted the changes in the manuscript using red sentences.

We hope that the new version of the manuscript has eliminated the weaknesses highlighted by this reviewer.

However, there are some issues that should be addressed, as detailed below.
Major comments:

1. While the study provided evidence for accumulation of SET protein affect neuronal specification and delayed neurodevelopment in in vitro and in vivo models, the authors made a strong statement that differential cis-regulation caused by elevated SET level as a key contributing factor for SET-related disorders including SGS, neurodevelopmental disorders, myeloproliferative diseases and cancer. However, as stated by the authors the Introduction and Discussion, "direct evidence of chromatic defects and their eventual connection has never been investigated." While it is an intriguing hypothesis, the authors might want to tune down the statement as the manuscript did not report evidence of differential cis-regulation in other SET-related disorders, e.g. cancer. Otherwise, citations of the relevant publications should be included and discussed.

We are aware of this, and we agree with this reviewer point. We intended suggesting that what we found in SGS could happen in other SET-related disorders. In the new version of the manuscript, we amended this misunderstanding with the following actions:

- tune down the claim in the abstract and in the text;
- the addition in different parts of the discussion of different remarks indicating that at this stage, this is the only study proposing this pathogenic mechanism linked with SETBP1 mutations;
- We underline especially for cancer that other mechanisms are well established as pathogenic;
- We stressed the need of further studies to clarify the extent and importance of this molecular function both in SGS and in SET-related diseases other than SGS.

2. While it is interesting to see that SET accumulation was shown in SGS NPCs, iPSCs and NPCs overexpressing SET and in zebrafish infected with SET mRNA, and that impairments in H3K27ac, chromatin accessibility and chromatin loop contacts have been shown in a combination of models, as the authors also pointed out in the Discussion that the contribution of SETBP1 cannot be overlooked, since it is a chromatin remodeller and transcription factor itself, which was also shown in the authors' previous publications. SETBP1 carrying an SGS variant is likely to affect chromatin functions and can cause these impairments, and the extent of contribution

of abnormal SETBP1 and SET level to these impairments is unknown. Without showing that reducing SETBP1 and SET protein level could reverse abnormal cis regulation reported, the author could not be sure that SETBP1 did not contribute to these impairments. The authors should therefore tune down their conclusion about SET and elaborate more on the plausible contribution of SGS SETBP1 in the Discussion. Otherwise, the authors should add an experiment addressing these points.

We understand the point of the reviewer. Indeed, we referred to our model as “[..]as **a contributing factor to the pathological basis [..]**” (abstract) and say “**the chromatin response to SET function as one of the possible pathological contributors to the diseases.**” in the text.

We agree that would be important try to strongly dissect the contribution of the two factors. In fact, the experimental setting we made regarding the sole upregulation of SET without neither SETBP1 accumulation nor SGS genotype (Extended Data Figure 1a-d) was intended in that direction, i.e. rule out the SETBP1-only effects. Although we demonstrated that the accumulation of SET seems to recapitulate the grossly some of the phenotypes of SGS patients derived cells, we acknowledge that the extensive work that we did on SGS models is missing on the “SET-only” model to fully support a definitive claim.

We also tried in this revision to perform knock-down experiments of SET to gain information on a possible reversal of the phenotype. Unfortunately, despite the fact that we achieved a good downregulation of SET (through shRNA), the manipulated cells did not support neural differentiation. We thus concluded that would need a long and dedicated process of study that is probably out of the scope of this work.

Said that, we agree with this reviewer to revise our conclusions and elaborate more on the contribution of SET-independent SETBP1 function in SGS.

Minor:

(The line number restarts on each page but there is no page number in the manuscript pdf so I will refer to the section and paragraph in the rest of the comments.)

Results section:

- In the second paragraph of “SET levels influence histone acetylation in multiple models”:

- o Line 7: it was not clear on which samples ChIP-seq with SET antibody was performed. Is there Was there differential binding targets of SET in different conditions or only the strength was different?

We have clarified better in the text that ChIP-seq for SET was performed only in SGS NPCs (control and mutant) and that the difference was identified at the level of SET binding strength.

- o In the following sentence in line 7-10, “At the genome-wide level, all SEThigh conditions examined were featured by consistently less called peaks compared to the relative controls (Extended Data Fig. 1i), while all pairwise comparisons were similar in term of accessibility of the identified peaks (Fig. 1g, i, Extended data Fig. 1j-k).”, it was not clear that it was referring to the ATAC-seq peak, please specify in the text.

ATAC-seq specification has been added in the text.

- In the second paragraph of “SGS NPCs display rearranged chromatin topography”, lines 23-26: “A similar pattern was observed in the RIMS4 locus (synaptic protein connected with autism cases). Notably, both genes resulted reduced in mutant cells in relation to the controls (Fig. 3i).” It should be (Extended Data Fig. 3g) instead of (Fig. 3i).

Corrected

- In the first paragraph of “Neuronal development is regulated in an alternative way in SGS”,

- o Line 8, “A straight comparison of open chromatin peaks between control NPCs... during neuronal differentiation in vitro (Fig. 4a).” This sentence is incomplete, rephrase.

Corrected

- o Line 12, “Anyway, at the qualitative level, the regions between the two genotypes were different, with only 4,488 and 2,796 peaks that gained and lost accessibility in both conditions, respectively (Fig. 4b, Extended Data Fig. 4c, Supplementary Table 9).”  it is unclear which “both conditions” were referred to, please specify.

Corrected

- In the second paragraph of “HDAC inhibitor fosters the maturation of SGS neurons”, line 9, “which successfully restored the H3 acetylation levels in SGS neurons (Extended Data Fig. 5a).”  but the figure showed data of SGS NPCs, please correct.

Corrected

- In the third paragraph of “HDAC inhibitor fosters the maturation of SGS neurons”, “These results sustain the model by which chromatin defects, secondary to high levels of SET, are responsible for defective neuronal differentiation in SGS.”  The authors cannot conclude from these data that high SET level is the only factor responsible for defective neuronal differentiation in SGS without an experiment reducing SET level and see if the phenotype was rescued, especially when the impairments were only alleviated.

We tuned down the sentence.

- The authors first showed that there was earlier opening of neuron-specific TFBS and persistent opening of TFBE typical to NPCs and persistent closure of neuronal-specific ones in SGS NPCs and neurons but delayed neuronal maturation was shown in SGS whole mouse brain. In the Discussion, it was also mentioned that there was early sign of degeneration with DNA damage in SGS organoids as shown in Banfi et al., 2021, are the mechanisms shown, i.e. delayed neuronal maturation and early degeneration of neurons mutually exclusive, or it is a combination of both? Please elaborate in the Discussion.

In our precedent work, we reported neuronal degeneration secondary to DNA damage. However, delayed maturation is present also in an experimental setting in which we bypass the DNA damage and thus the neurodegeneration. At this moment we cannot exclude that, in the context of the patients, these two elements (delay in maturation and neurodegeneration) may act in combination. We underlined this in the new version of the discussion.

- Legends of Fig 4h, the figure showed data of SEMA3A but legends said RSPH3.

Corrected

- Legends of Fig 6l, should be E14.5 instead of E24.5

Corrected

- Legends of Extended Data Fig. 6e, “CMV::Cre and Rosa26-LoxP-STOP-LoxP-hSETBP1G870S” should be “Nestin::Cre” if it is for getting whole brain specific mutant?

Corrected

- Legends of Extended Data Fig. 6g, E14.5 instead of E145.

Corrected

REVIEWERS' COMMENTS

Reviewer #2 (Remarks to the Author):

I appreciate the additional analyses and data the authors provided in response to my initial comments. They improve the quality of the study and reaffirm the conclusion of the study. I fully support the publication of this work.

Reviewer #3 (Remarks to the Author):

The authors have satisfactorily addressed my comments by clarifications and additional discussion in the main text. The authors tried to perform additional knock-down experiments of SET to gain information on a possible reversal of the phenotype but unfortunately, the cells where SET was downregulated through shRNA could not differentiate into neural fate. Could it be a technical issue or biologically relevant since the balance of SET dosage is important for neural differentiation? If it is the latter, could the authors include this preliminary data in the supplementary information? If it is technical issue, I agree with the authors that this would need a long and dedicated process that is out of the scope of this work.

There is a minor typo in the legend in Fig 4, last word SEMA3A instead of RSPH3.

Overall, I think the additional analysis and the clarifications added to the main text have improved the manuscript. I would like to congratulate them for a novel, thorough and comprehensive work.

Point by Point Response about our MS NCOMMS-22-43699A by Zaghi et al.

Reviewer #2 (Remarks to the Author):

I appreciate the additional analyses and data the authors provided in response to my initial comments. They improve the quality of the study and reaffirm the conclusion of the study. I fully support the publication of this work.

We really thank the reviewer for her/his generous support.

Reviewer #3 (Remarks to the Author):

The authors have satisfactorily addressed my comments by clarifications and additional discussion in the main text. The authors tried to perform additional knock-down experiments of SET to gain information on a possible reversal of the phenotype but unfortunately, the cells where SET was downregulated through shRNA could not differentiate into neural fate. Could it be a technical issue or biologically relevant since the balance of SET dosage is important for neural differentiation? If it is the latter, could the authors include this preliminary data in the supplementary information? If it is technical issue, I agree with the authors that this would need a long and dedicated process that is out of the scope of this work.

As the reviewer correctly said we have tried to perform SET knock-down experiments. We have reached good downregulation of SET through the usage of specific shRNA. The KD NPCs failed to differentiate in neurons, dying in the plate. Despite we suspect that this may have biological relevance, at this point we do not have enough evidence to strongly claim this. Indeed, we have to replicate the experiments both in other lines and by other means (e.g. KO by CRISPR). We also have to check properly off-targets and make rescue experiments with KD resistant transgene. We hope that the reviewer would agree with us that such a strong claim on this basis is inappropriate in this context.

There is a minor typo in the legend in Fig 4, last word SEMA3A instead of RSPH3.

Corrected, thank you

Overall, I think the additional analysis and the clarifications added to the main text have improved the manuscript. I would like to congratulate them for a novel, thorough and comprehensive work.

We really thank the reviewer for her/his generous support.